# Oct4 differentially regulates chromatin opening and enhancer transcription in pluripotent stem cells

Le Xiong[1†], Erik A Tolen[2†], Jinmi Choi[1], Sergiy Velychko[2], Livia Caizzi[1], Taras Velychko[1], Kenjiro Adachi[2], Caitlin M MacCarthy[2], Michael Lidschreiber[1], Patrick Cramer[1]*, Hans R Schöler[2]*

[1]Max Planck Institute for Multidisciplinary Sciences, Department of Molecular Biology, Göttingen, Germany; [2]Max Planck Institute for Molecular Biomedicine, Department of Cell and Developmental Biology, Münster, Germany

**Abstract** The transcription factor Oct4 is essential for the maintenance and induction of stem cell pluripotency, but its functional roles are not fully understood. Here, we investigate the functions of Oct4 by depleting and subsequently recovering it in mouse embryonic stem cells (ESCs) and conducting a time-resolved multiomics analysis. Oct4 depletion leads to an immediate loss of its binding to enhancers, accompanied by a decrease in mRNA synthesis from its target genes that are part of the transcriptional network that maintains pluripotency. Gradual decrease of Oct4 binding to enhancers does not immediately change the chromatin accessibility but reduces transcription of enhancers. Conversely, partial recovery of Oct4 expression results in a rapid increase in chromatin accessibility, whereas enhancer transcription does not fully recover. These results indicate different concentration-dependent activities of Oct4. Whereas normal ESC levels of Oct4 are required for transcription of pluripotency enhancers, low levels of Oct4 are sufficient to retain chromatin accessibility, likely together with other factors such as Sox2.

*For correspondence:
patrick.cramer@mpibpc.mpg.
de (PC);
h.schoeler@mpi-muenster.mpg.
de (HRS)

†These authors contributed
equally to this work

Competing interest: The authors declare that no competing interests exist.

## Editor's evaluation

The manuscript by Xiong et al. provides high-resolution kinetic information on transcriptional events and enhancer activity after the loss of the pluripotency factor Oct4. The authors demonstrate different concentration-dependent activities of Oct4 in controlling enhancer transcription and chromatin accessibility. These results are of interest to stem cell biologists and developmental biologists.

## Introduction

The transcription factor (TF) Oct4 is essential for maintaining pluripotency in vitro (*Niwa et al., 2000*) as well as in vivo (*Nichols et al., 1998*). Oct4 lies at the core of an intricate transcriptional regulatory network that maintains the pluripotent state (*Boyer et al., 2005*; *Zhou et al., 2007*) and Oct4-driven protein–protein interactions are crucial for maintaining and inducing pluripotency (*Esch et al., 2013*; *Han et al., 2022*; *van den Berg et al., 2010*). Oct4 knockout in mouse embryonic stem cells (ESCs) leads to a quick disruption of the pluripotency state and differentiation of the ESCs into trophecto-derm (*Niwa et al., 2000*). Oct4 binds to enhancers (*Schöler et al., 1989*), which are cis-regulatory genomic elements that orchestrate gene expression in metazoans (*Banerji et al., 1981*). Oct4 function in maintaining pluripotency of ESCs has been attributed to the establishment of superenhancers (SEs), that have high occupancy of TFs and coactivators (*Hnisz et al., 2013*; *Whyte et al., 2013*). Degradation of Oct4 leads to preferential decrease of Oct4 and Mediator occupancy at SEs (*Boija*

*et al., 2018*) and consequent downregulation of pluripotency genes that are located near SEs (*Whyte et al., 2013*).

Oct4 cooperates with the TF Sox2 at thousands of genomic sites in ESCs (*Ambrosetti et al., 1997*; *Chen et al., 2008*; *Loh et al., 2006*; *Yuan et al., 1995*). Oct4 and Sox2 cooperatively bind the SoxOct composite motif present in their target enhancers (*Chen et al., 2008*; *Lam et al., 2012*). A previous study showed that prolonged depletion of Oct4 for 24 hr in ESCs resulted in a loss of chromatin accessibility at the majority of its occupied sites, accompanied by a loss of Sox2 binding (*King and Klose, 2017*). Using an Oct4 protein depletion system in ESCs, another study suggested that chromatin accessibility is regulated in a highly dynamic manner with Oct4 being constantly required (*Friman et al., 2019*). However, enhancer-driven gene expression regulation involves both the control of enhancer activity and chromatin accessibility, and it remains unclear how the two processes are regulated by Oct4 and how this amounts to the control of gene transcription.

Enhancers are often transcribed resulting in the production of enhancer RNAs (eRNAs) (*De Santa et al., 2010*; *Kim et al., 2010*; *Tuan et al., 1992*). The functions of enhancer transcription and eRNA remain poorly understood (*Lewis et al., 2019*; *Li et al., 2016*; *Sartorelli and Lauberth, 2020*). Transcribing Pol II at enhancers contributes to chromatin alterations by recruiting histone modifying and remodeling factors (*Ho et al., 2006*; *Kaikkonen et al., 2013*; *Ling et al., 2004*), and eRNA may be involved in gene regulation (*Bose et al., 2017*; *Gorbovytska et al., 2021*; *Kaikkonen et al., 2013*; *Li et al., 2013*; *Mousavi et al., 2013*; *Schaukowitch et al., 2014*; *Sigova et al., 2015*). The synthesis of eRNA correlates with the transactivation activity of enhancers (*Andersson et al., 2014*; *De Santa et al., 2010*; *Djebali et al., 2012*; *Hah et al., 2013*; *Henriques et al., 2018*; *Kim et al., 2010*; *Melgar et al., 2011*; *Michel et al., 2017*; *Wu et al., 2014*). The synthesis of eRNA can be used to identify putative enhancers by transient transcriptome sequencing (TT-seq), a method that captures newly synthesized RNA (*Schwalb et al., 2016*). TT-seq combines a short pulse of 4-thiouridine (4sU) labeling with RNA fragmentation and monitors transcription changes at both enhancers and their target genes genome-wide (*Schwalb et al., 2016*). TT-seq can quantify changes in enhancer transcription and is ideal to monitor immediate transcriptome changes after perturbation (*Choi et al., 2021*; *Gressel et al., 2019*; *Michel et al., 2017*).

To investigate the functional roles of Oct4 in the control of pluripotency, we used depletion and subsequent recovery of Oct4 in mouse ESCs. We then conducted a high-resolution time course study to monitor changes in the transcriptome by TT-seq, changes in chromatin accessibility by ATAC-seq (*Buenrostro et al., 2013*), and changes in Oct4 and Sox2 occupancy by ChIP-seq. During Oct4 depletion, we found that loss of Oct4 from enhancers goes along with a decrease in mRNA synthesis from Oct4 target genes crucial for maintaining pluripotency. Moreover, eRNA synthesis decreased rapidly at Oct4-bound enhancers, whereas chromatin accessibility was generally decreased only later, after Oct4 levels had considerably dropped. In contrast, during partial recovery of Oct4 after full depletion, chromatin accessibility restored rapidly, whereas eRNA synthesis could not be fully recovered. These results suggest that low levels of Oct4 are sufficient to regulate chromatin accessibility, whereas normal ESC levels of Oct4 are required to maintain the transcription of pluripotency enhancers.

## Results
### Oct4 depletion and transcription unit annotation
To investigate the role of Oct4 in maintaining pluripotency, we used a doxycycline (DOX) inducible Oct4 knockout mouse embryonic stem cell line (mESC) ZHBTc4 (*Niwa et al., 2000*). This system was previously used to study the effect of Oct4 depletion after 24 hr (*Friman et al., 2019*; *King and Klose, 2017*). To investigate the direct, primary role of Oct4, we conducted a time course experiment collecting samples after 0, 3, 6, 9, 12, and 15 hr of DOX treatment. We found that Oct4 protein levels were already reduced after 6 hr of treatment and substantially decreased after 9 hr (*Figure 1A*, whole cell lysate). Oct4 protein depletion was complete at 24 hr of treatment, while Sox2 and Nanog protein levels remained mostly unchanged for extended times before eventually decreasing (*Figure 1A*, *Figure 1—figure supplement 1A*). Chromatin binding of Oct4 was reduced already after 3 hr of DOX treatment, whereas Sox2 binding decreased after 9 hr and Nanog binding remained unchanged over the entire time course (*Figure 1A*, chromatin).

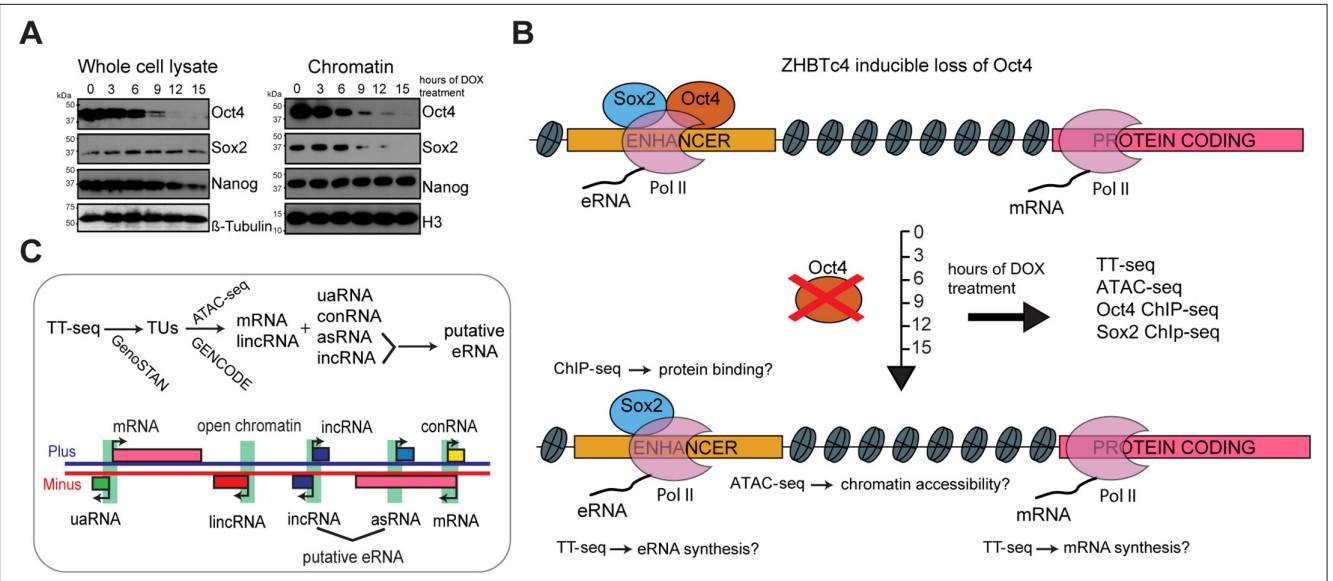

**Figure 1.** Oct4 depletion in ZHBTc4 mouse embryonic stem cells (ESCs). (**A**) Western blot analysis of whole cell lysate and chromatin samples over the time course of doxycycline (DOX) treatment using Oct4, Sox2, Nanog, β-Tubulin (control), and H3 (control) antibodies. In the chromatin fraction Oct4 levels gradually decreased to 87.5% (3 hr), 50.5% (6 hr), 17.0% (9 hr), 7.0% (12 hr), and 2.5% (15 hr), whereas Sox2 levels slightly increased to 126.8% (3 hr) and 112.5% (6 hr), and then decreased to 32.4% (9 hr), 16.4% (12 hr), and 1.7% (15 hr) compared to 0 hr. (**B**) Schematic of methodology and samples collected. TT-seq, ATAC-seq, Oct4, and Sox2 ChIP-seq experiments were performed after 0, 3, 6, 9, 12, and 15 hr of DOX treatment in ZHBTc4 mouse ESCs. (**C**) Transcription unit (TU) annotation. Genome segmentation with GenoSTAN was used to annotate TUs from TT-seq data. ATAC-seq data and mouse GENCODE annotation were then used to classify TUs.

The online version of this article includes the following source data and figure supplement(s) for figure 1:

**Source data 1.** Source data for *Figure 1A*.

**Source data 2.** Source data for *Figure 1A*.

**Figure supplement 1.** Transcription unit annotation in ZHBTc4 mouse ES cell line, related to *Figure 1*.

To monitor the effect of a rapid Oct4 depletion on transcription, we conducted TT-seq (*Schwalb et al., 2016*) after 0, 3, 6, 9, 12, and 15 hr of DOX treatment (*Figure 1B*). RNA labeling with 4sU was carried out for 5 min and two independent biological replicates were generated for each time point (*Supplementary file 1*). To study the role of Oct4 in maintaining chromatin accessibility, we performed ATAC-seq (*Buenrostro et al., 2013*) over the same time course (*Figure 1B*, *Supplementary file 2*). TT-seq and ATAC-seq data were highly reproducible (*Figure 1—figure supplement 1B, C*).

We then used the TT-seq data to segment the genome into transcription units (TUs) and nontranscribed regions using GenoSTAN (*Zacher et al., 2017*; *Figure 1C* and Materials and methods). To avoid spurious predictions, TUs detected by TT-seq had to exceed a minimal expression cutoff of RPK >26.5 and had to originate from an open chromatin region identified by ATAC-seq (*Figure 1—figure supplement 1D, E*). We sorted TUs into protein-coding RNAs (mRNAs) and long intergenic noncoding RNAs (lincRNAs) based on GENCODE annotation (*Frankish et al., 2019*). The remaining noncoding TUs were classified as upstream antisense RNA (uaRNA), convergent RNA (conRNA), antisense RNA (asRNA), and intergenic RNA (incRNA) units according to the location relative to mRNA (*Figure 1C* and Materials and methods). This resulted in 9266 mRNAs, 9257 incRNAs, 3661 asRNAs, 1981 uaRNAs, 471 conRNAs, and 318 lincRNAs (*Figure 1—figure supplement 1F*). The length distribution of the detected RNA units (*Figure 1—figure supplement 1G*) agreed with previous estimations (*Michel et al., 2017*; *Schwalb et al., 2016*).

## Oct4 maintains the transcriptional network governing pluripotency

We first investigated changes in mRNA synthesis during Oct4 depletion. Changes could already be observed after 3 hr of DOX treatment (*Figure 2—figure supplement 1A*), in agreement with chromatin fractionation results (*Figure 1A*, **chromatin**). Differential gene expression analysis (*Love et al., 2014*) detected 769 significantly downregulated and 829 upregulated genes (adjusted p value <0.01)

after 15 hr of DOX treatment (*Figure 2A, B*, *Figure 2—figure supplement 1B*). To dissect the kinetics of mRNA synthesis changes of differentially expressed genes, we performed *k*-means clustering and classified early and late down- and upregulated genes (*Supplementary file 3*). Early downregulated genes (446 genes) showed the strongest decrease in mRNA synthesis after 6–9 hr of DOX treatment (*Figure 2C, D*, *Figure 2—figure supplement 1C*, left), whereas late downregulated genes (323 genes) decreased most strongly after 12–15 hr (*Figure 2C, D*, *Figure 2—figure supplement 1C*, right). Early and late upregulated genes behaved similarly (*Figure 2—figure supplement 1D–F*). Gene ontology (GO) analysis (*Huang et al., 2009*) showed that early downregulated genes were enriched for stem cell population maintenance (*Figure 2E*), whereas late downregulated genes were enriched for DNA replication and cell cycle (*Figure 2—figure supplement 1G*). Early upregulated genes showed enrichment for carbohydrate metabolic processes (*Figure 2—figure supplement 1H*), and late upregulated genes were enriched for in utero embryonic development (*Figure 2—figure supplement 1I*). These findings reflect the differentiation of ESCs into trophectoderm upon loss of Oct4 (*Niwa et al., 2000*).

We then assessed if there was an enrichment of putative SE-controlled genes among the early downregulated genes. Indeed, of the 150 transcribed genes that are nearest to SEs (*Whyte et al., 2013*), 60 were significantly downregulated, of which 45 were early downregulated (*Figure 2F*, p value = 2.6e−24, Fisher's exact test). We then compared the kinetics of mRNA synthesis changes of the putatively SE-controlled downregulated genes (60) to other downregulated genes (709). mRNA synthesis of SE-controlled downregulated genes was particularly sensitive to Oct4 depletion (*Figure 2G*). Among the 60 SE genes that were downregulated, we found many pluripotency genes at early time points (*Figure 2H*). At 6 hr of Oct4 depletion we found a significant downregulation of *Esrrb*, *Klf2*, *Klf4*, *Utf1*, and *Tbx3*. At 9 hr of depletion, *Sox2*, *Nanog*, and *Prdm14* were significantly downregulated, and *Nr5a2* and *Fgf4* after 12 hr. Taken together, our analysis of early mRNA synthesis changes upon Oct4 depletion revealed a rapid downregulation of the components of the pluripotency transcriptional network with SE-controlled genes being immediately and strongly affected. Thus, consistent with previous findings (*Whyte et al., 2013*), Oct4 is required to maintain the transcriptional network underlying pluripotency.

## Oct4-bound transcribed enhancers produce high levels of eRNAs

To understand how loss of Oct4 leads to rapid destabilization of the pluripotency gene network, we combined our TT- and ATAC-seq data with published Oct4 ChIP-seq data (*Supplementary file 4*; *King and Klose, 2017*) to annotate putative enhancers in mESCs (*Figure 3A*). First, we defined transcribed enhancers by annotating putative eRNAs (*Figure 3A*). We selected asRNAs and incRNAs that originated over 1 kb away from promoter-related RNAs (mRNA, conRNA, and uaRNA) and merged those located less than 1 kb apart (*Figure 1C*). This resulted in 8727 putative eRNAs, consisting of 2468 asRNAs and 6259 incRNAs, with a median length of ~700 bp (*Figure 3—figure supplement 1A, B*). Most Oct4 ChIP-seq peaks (91%) overlapped with open chromatin regions identified by ATAC-seq (*Figure 3—figure supplement 1C*). Out of the 8727 putative eRNAs, 2221 overlapped with 2231 Oct4-bound sites (Klf4 SE shown as an example in *Figure 3—figure supplement 1D*). We refer to these Oct4-bound sites as Oct4-bound transcribed enhancers (*Figure 3A*). The majority of Oct4-bound transcribed enhancers (90%) were marked by histone H3 lysine 4 mono-methylation (H3K4me1) (*Figure 3—figure supplement 1E*). Oct4-bound transcribed enhancers were strongly enriched for biological processes related to stem cell population maintenance (*Figure 3—figure supplement 1F, G*). eRNAs originating from Oct4-bound enhancers were significantly longer and showed higher synthesis than other eRNAs (*Figure 3—figure supplement 1E*, p value <2.2e−16, Wilcoxon rank sum test).

Of the remaining Oct4-bound accessible sites, 1098 produced mRNAs, and 12,710 produced no detectable RNAs and were referred to as Oct4-bound nontranscribed enhancers (*Figure 3A, B*). We then performed metagene analysis for TF enrichment at Oct4-bound transcribed and nontranscribed enhancers using published data (*Chronis et al., 2017*; *King and Klose, 2017*; *Supplementary file 4*). Whereas both groups of enhancers showed similar H3K4me1 levels, Oct4-bound transcribed enhancers were enriched with active histone marks H3K27ac and H3K4me3, higher chromatin accessibility, and higher occupancies of Oct4, Sox2, Nanog, Klf4, and Esrrb (*Figure 3C*, *Figure 3—figure supplement 1H*). Moreover, Oct4-bound transcribed enhancers were located closer to their nearest active putative target genes (median distance of 37 kb) as compared to nontranscribed enhancers

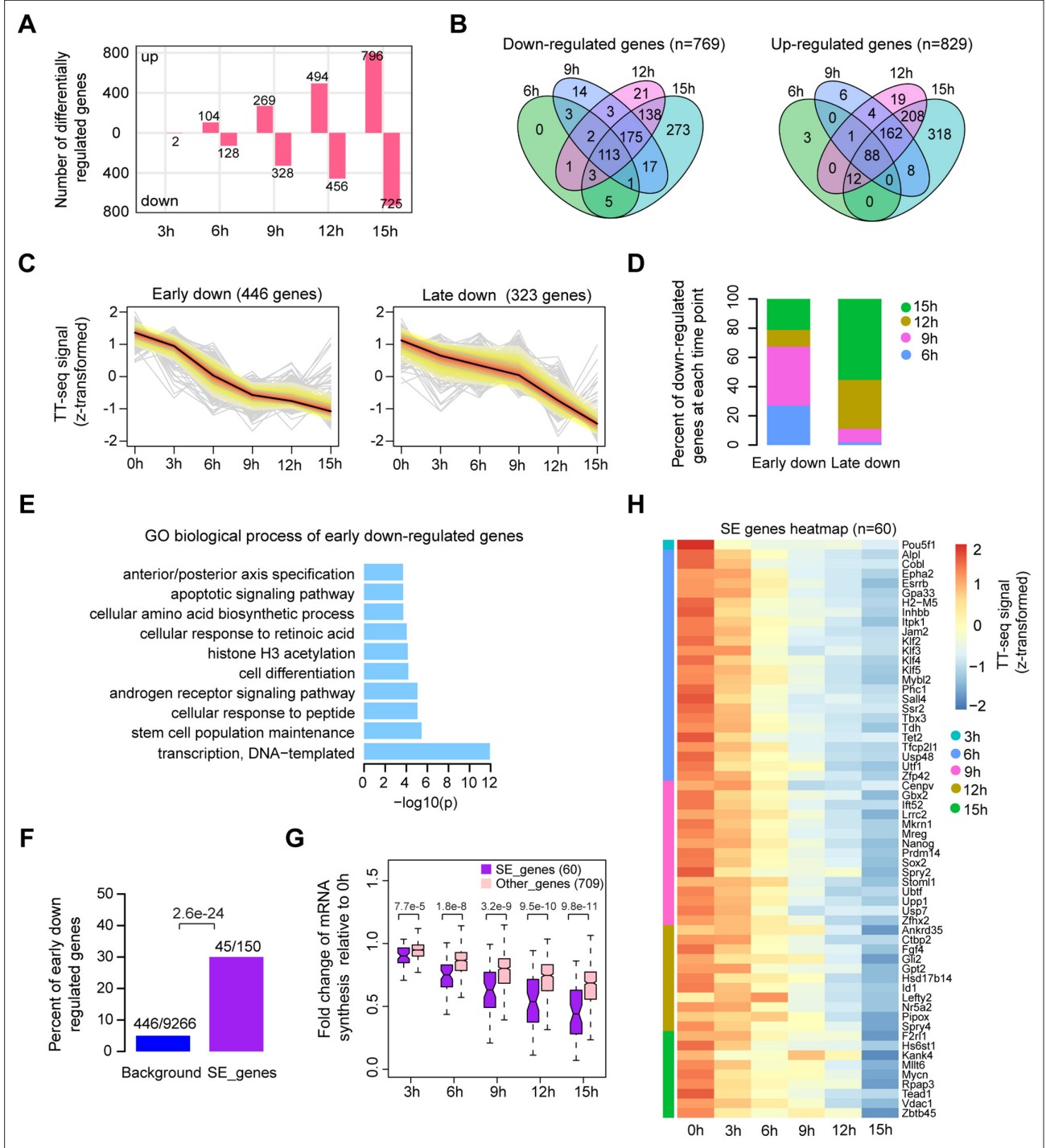

**Figure 2.** Oct4 maintains the transcriptional network governing pluripotency. (**A**) The number of differentially expressed mRNAs detected by DESeq2 after each time point of doxycycline (DOX) treatment. (**B**) Venn diagram showing the overlap for differentially expressed mRNAs detected by DESeq2 at each time point of DOX treatment. (**C**) *k*-Means clustering of 769 downregulated mRNAs into early (left) and late (right) downregulated groups. *y*-Axis indicates *z*-score transformed TT-seq counts. (**D**) Differentially regulated time point composition for early (left) and late (right) downregulated gene groups. *y*-Axis indicates percentage. (**E**) Gene ontology (GO) biological process enrichment of early downregulated mRNAs. (**F**) Bar graph depicting the percentage of early downregulated mRNAs in all annotated mRNAs (blue, as background) and SE-controlled mRNAs (purple). Superenhancer (SE) annotation was obtained from *Whyte et al., 2013*. p value was calculated by Fisher's exact test. (**G**) Boxplot indicating the changes in mRNA synthesis for putative SE-controlled downregulated genes (*n* = 60, purple) versus other downregulated genes (*n* = 709, pink). *y*-Axis indicates fold change of mRNA synthesis relative to 0 hr. p values were calculated by Wilcoxon rank sum test. Black bars represent the median values for each group. Lower and upper boxes are the first and third quartiles, respectively. The ends of the whiskers extend the box by 1.5 times the interquartile range. Outliers are not

*Figure 2 continued*

drawn. (**H**) Heatmap indicating the kinetics of SE-controlled downregulated genes (n = 60). Genes were ordered by the corresponding time of significant downregulation. Note that data from two biological replicates were generated for TT-seq and that the two replicates were merged for illustration.

The online version of this article includes the following source data and figure supplement(s) for figure 2:

**Source data 1.** Source data for *Figure 2*.

**Figure supplement 1.** Differential gene expression analysis and clustering, related to *Figure 2*.

(median distance 89 kb) (*Figure 3—figure supplement 1I*). Finally, we investigated eRNA synthesis at SEs. Half of the Oct4-bound sites within SEs produced eRNAs (*Figure 3D*). The eRNAs obtained from SEs were generally longer and had higher synthesis levels compared to eRNAs from typical enhancers (TEs), and SEs showed higher occupancy with Oct4, Sox2, Nanog, Klf4, and Esrrb (*Figure 3—figure supplement 1J, K*). These efforts led to a refined enhancer annotation in mESCs and suggest that Oct4-bound enhancers are transcriptionally more active than other enhancers.

## Oct4 is often required for enhancer transcription

We next analyzed Oct4-bound transcribed enhancers (*Figure 3A*) with respect to changes in their eRNA synthesis upon Oct4 depletion. Synthesis of eRNAs was highly reproducible between the two biological replicates (*Figure 4—figure supplement 1A*). Principle component analysis (PCA) revealed that the changes of Oct4-regulated eRNA synthesis followed a similar trajectory to that seen for mRNAs (*Figure 4—figure supplement 1B*, *Figure 2—figure supplement 1A*). Differential expression

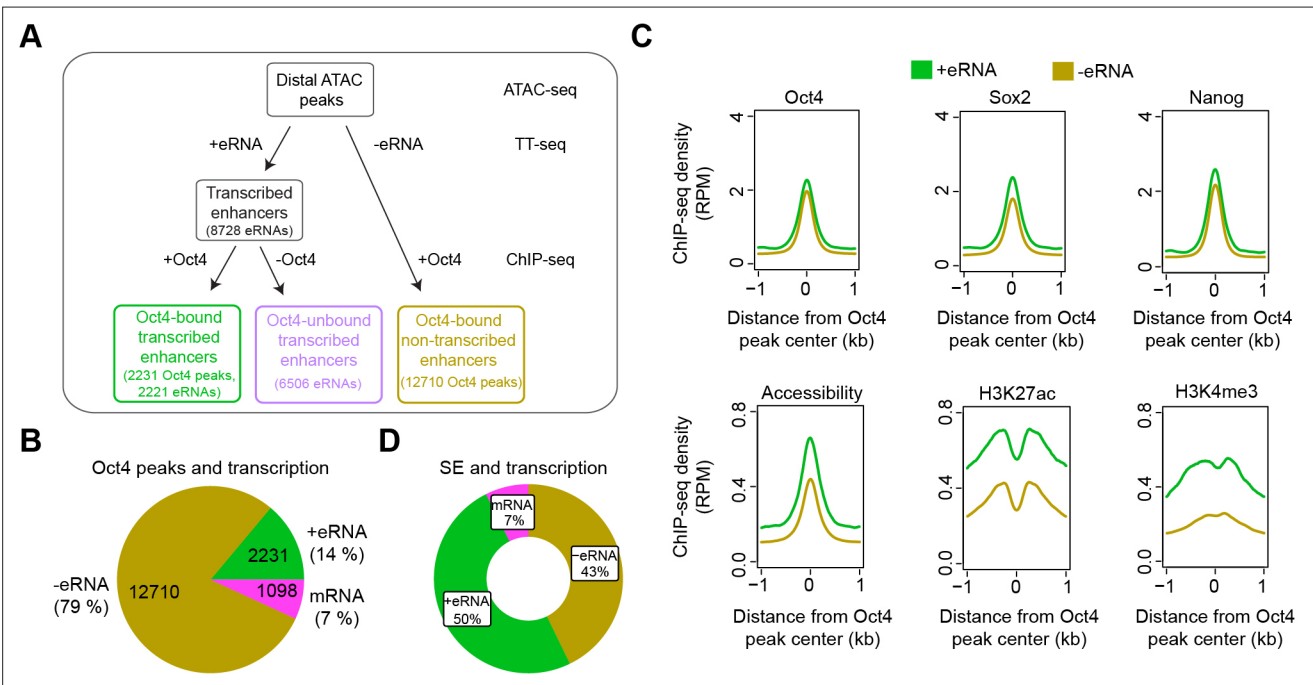

**Figure 3.** Annotation of putative Oct4-bound/regulated enhancer classes in mouse embryonic stem cells (mESCs). (**A**) Diagram illustrating classification of Oct4-binding sites by combing ATAC-seq, TT-seq annotated eRNAs, and Oct4 ChIP-seq peaks. Oct4 ChIP-seq data at 0 hr doxycycline (DOX) treatment were obtained from *King and Klose, 2017*. (**B**) Pie chart indicating the overlap of Oct4-binding sites with regions of active transcription (eRNA or mRNA) or no transcription annotated by TT-seq. (**C**) Metagene plot showing the occupancy for transcription factors Oct4, Sox2, Nanog, chromatin accessibility, and H3K27ac and H3K4me3 histone modifications at Oct4-bound transcribed enhancers (n = 2231) and Oct4-bound nontranscribed enhancers (n = 12,710). y-Axis depicts ChIP-seq coverage density in reads per million (RPM). ChIP-seq data of Oct4, Sox2, and Nanog were obtained from *King and Klose, 2017*, H3K27ac and H3K4me3 histone modifications data were obtained from *Chronis et al., 2017*. (**D**) Pie chart depicting the overlap of Oct4-bound sites (n = 514) at superenhancer (SE) that show eRNA transcription (n = 256, 50%), mRNA transcription (n = 38, 7%), or no transcription (n = 220, 43%) by TT-seq.

The online version of this article includes the following figure supplement(s) for figure 3:

**Figure supplement 1.** Annotation of eRNA and characterization of Oct4-bound transcribed enhancers in mouse embryonic stem cell (mESC), related to *Figure 3*.

analysis of eRNAs (*Love et al., 2014*) detected significant downregulation of 782 Oct4-regulated eRNAs after 15 hr of DOX treatment (*Figure 4A*, *Figure 4—figure supplement 1C*, adjusted p value <0.01). The kinetics analysis showed that for the downregulated eRNAs synthesis decreased already after 3 hr (*Figure 4B*). Moreover, SE eRNAs were more strongly downregulated compared to TE eRNAs (*Figure 4C, D*). As a control, we also performed differential expression analysis of eRNAs for Oct4-unbound transcribed enhancers (*Figure 4—figure supplement 2A–E*, adjusted p value <0.01). Compared to Oct4-bound transcribed enhancers, the proportion of downregulated eRNAs was ~five-fold lower for Oct4-unbound transcribed enhancers, and their eRNA synthesis changes were generally observed later, likely representing secondary effects (*Figure 4—figure supplement 2F*). Taken together, these results suggest that Oct4 is required for eRNA synthesis at about one third of putative Oct4-bound transcribed enhancers including SEs.

## Oct4 binds enhancers to activate putative target genes

To investigate whether Oct4 depletion leads to a loss of Oct4 binding to enhancers, we performed ChIP-seq of Oct4 after 0, 3, 6, 9, 12, and 15 hr of DOX treatment (*Supplementary file 5*). In agreement with chromatin fractionation results (*Figure 1A*, **chromatin**), Oct4 occupancy decreased after 3 hr and strongly dropped after 9 hr of DOX treatment (*Figure 4E*). This is consistent with the observed decrease in eRNA synthesis (*Figure 4B*). In accordance with our previous results (*Figure 4D*), occupancy of Oct4 decreased more strongly at SEs compared to TEs (*Figure 4F*). These results show that Oct4 binding is required for eRNA synthesis at a subset of Oct4-bound transcribed enhancers and particularly at SEs.

To investigate whether the observed decrease in Oct4 occupancy and eRNA synthesis at Oct4-bound transcribed enhancers coincided with a decrease of target mRNA synthesis, we paired transcriptionally downregulated SEs with their nearest transcribed genes. About half of these genes were downregulated (62 enhancer–gene pairs), and we found that mRNA synthesis decreased already after 3 hr of DOX treatment (*Figure 4G*). This is illustrated for the *Klf4* and *Sox2* genes (*Figure 4H, I*). For the remaining half of SE-nearest genes mRNA synthesis was unaffected. In addition to the possibility that the target gene may not always be the nearest gene, we found significantly lower occupancy of Oct4 (*Figure 4—figure supplement 3A*, left) as well as significantly lower decrease of eRNA synthesis (*Figure 4—figure supplement 3A*, right) in comparison to the affected SEs. This suggests that Oct4 occupancy and the degree of eRNA synthesis changes at transcriptionally downregulated SEs may play a role in transcription. Overall, these results are consistent with a function of Oct4 in activating both enhancer transcription and mRNA synthesis from its target genes.

## Reduced Oct4 binding does not immediately influence chromatin accessibility

We next investigated changes in chromatin accessibility at Oct4-bound transcribed enhancers. PCA indicated that accessibility changes started to occur after 6 hr of DOX treatment, with the most substantial changes happening before 12 hr (*Figure 5—figure supplement 1A*). To call significantly changed accessible chromatin regions we used DESeq2 (*Love et al., 2014*). In contrast to mRNAs and eRNAs (*Figures 2A and 4A*), only few enhancers were detected to have significantly altered chromatin accessibility at 6 hr (*Figure 5A*, *Figure 5—figure supplement 1B*). 15 hr of DOX treatment resulted in a significant decrease of chromatin accessibility at 726 enhancers (adjusted p value <0.01) (*Figure 5A*, *Figure 5—figure supplement 1B*). The kinetic analysis showed that for these enhancers, chromatin accessibility remained largely unchanged at 3 hr (*Figure 5B*). Moreover, for the downregulated SE–gene pairs chromatin accessibility was unaffected at 3 hr (*Figure 4G–I*). These results show that decreased Oct4 binding does not immediately lead to chromatin accessibility changes, which occur later into the time course when Oct4 levels had considerably dropped.

At Oct4-unbound transcribed enhancers chromatin accessibility remained largely unaltered during Oct4 depletion (*Figure 5—figure supplement 2*). To further investigate the role of Oct4 in altering chromatin, we classified the downregulated Oct4-bound transcribed enhancers (*Figure 4B*) based on their respective changes in chromatin accessibility. We defined two groups of downregulated transcribed enhancers, showing either decreased or unchanged chromatin accessibility (*Figure 5C*). In both groups, Oct4 binding to chromatin decreased over the time course and was associated with a decrease in eRNA synthesis (*Figure 5D, E*). For the first group, depletion of Oct4 led to a decrease

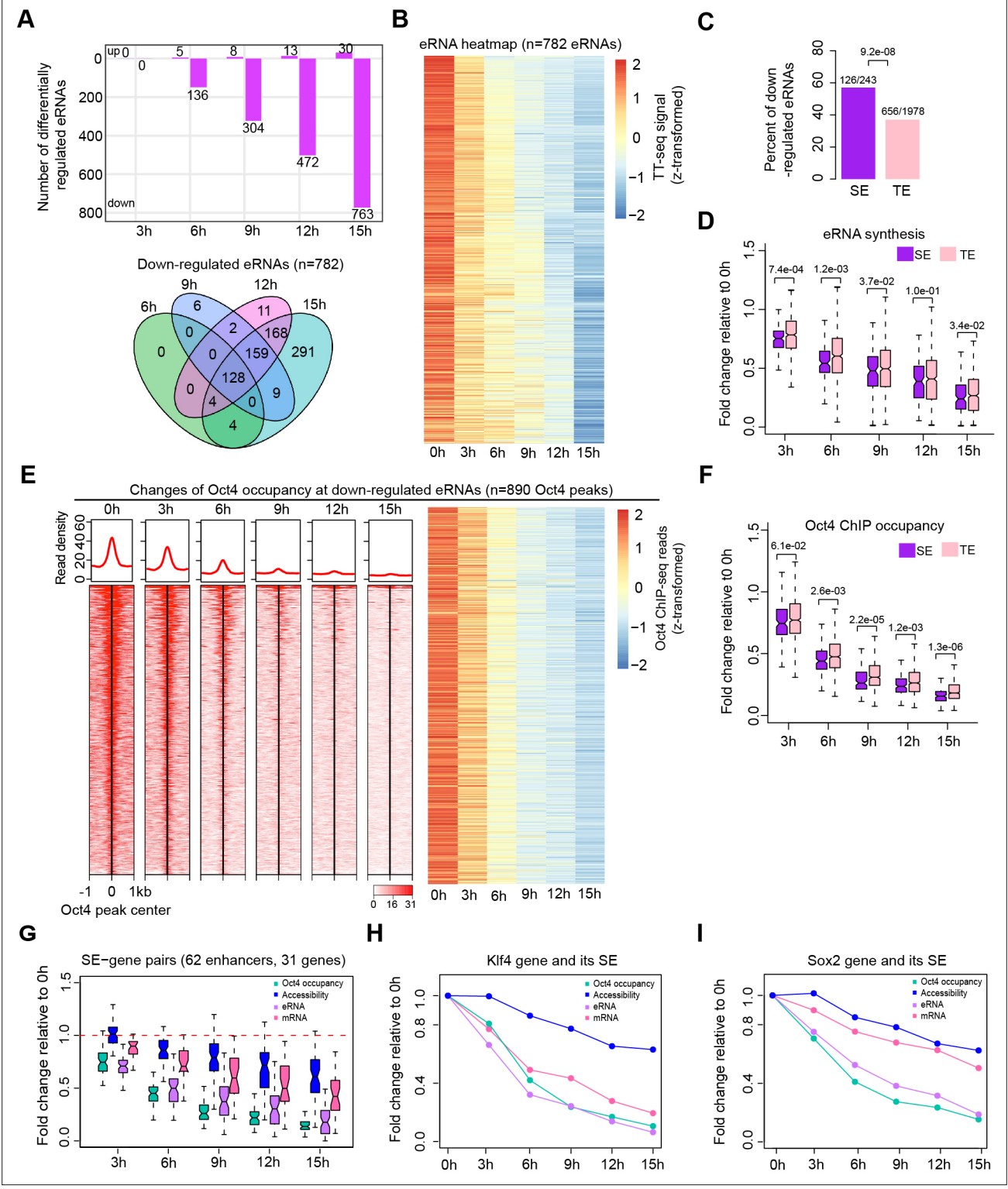

**Figure 4.** Oct4 is required for enhancer and gene transcription. (**A**) The number of differentially expressed Oct4-regulated eRNAs detected by DESeq2 (top) and Venn diagram showing overlapping of differentially expressed Oct4-regulated eRNA (bottom) at each time point of doxycycline (DOX) treatment. (**B**) Heatmap visualizing the kinetics of Oct4-regulated downregulated eRNAs (*n* = 782). (**C**) Bar chart indicating the percentage of downregulated eRNAs at superenhancer (SE) and typical enhancer (TE). p value was calculated by Fisher's exact test. (**D**) Boxplot indicating synthesis changes of downregulated eRNAs at SE (*n* = 126) versus TE (*n* = 656). *y*-Axis indicates fold change of eRNAs synthesis relative to 0 hr. p values were calculated by Wilcoxon rank sum test. Black bars represent the median values for each group. Lower and upper boxes are the first and third quartiles,

*Figure 4 continued on next page*

*Figure 4 continued*

respectively. The ends of the whiskers extend the box by 1.5 times the interquartile range. Outliers are not drawn. (**E**) Changes of Oct4 occupancy at downregulated eRNAs, illustrated by Oct4 ChIP-seq coverage (left) and count heatmaps (right). 782 downregulated eRNAs originated from 890 Oct4-bound transcribed enhancers (peaks). Normalized read densities are shown and peaks were ranked accordingly. (**F**) Boxplot indicating the corresponding Oct4 occupancy changes at downregulated SE versus TE eRNAs defined in (**D**). *y*-Axis indicates fold change of Oct4 occupancy relative to 0 hr. p values were calculated by Wilcoxon rank sum test. (**G**) Boxplot indicating changes of Oct4 occupancy, chromatin accessibility, eRNA and mRNA synthesis for 62 SE–gene pairs. *y*-Axis indicates fold change relative to 0 hr. Transcriptionally, downregulated SEs were paired with their nearest transcribed genes and pairs with downregulated genes were kept. Signal for individual Oct4-bound transcribed enhancers within SEs was plotted. (**H**) Fold changes of Oct4 occupancy, chromatin accessibility, eRNA and mRNA synthesis at the *Klf4* gene and its associated SE. Fold changes of Oct4 occupancy, chromatin accessibility, and eRNA synthesis at the SE were represented by the average of the three individual enhancers within the SE (illustrated as IGV track in *Figure 5F*). (**I**) Fold changes of Oct4 occupancy, chromatin accessibility, eRNA and mRNA synthesis at the *Sox2* gene and its associated SE. Fold changes of Oct4 occupancy and eRNA synthesis at the SE were represented by the average of the three individual enhancers within the SE (illustrated as IGV track in *Figure 5G*). Note that data from two biological replicates were generated for all assays and that the two replicates were merged for illustration.

The online version of this article includes the following figure supplement(s) for figure 4:

**Figure supplement 1.** eRNA synthesis changes at Oct4-bound transcribed enhancers, related to *Figure 4*.

**Figure supplement 2.** Differential expression analysis for eRNAs at Oct4-unbound transcribed enhancers, related to *Figure 4*.

**Figure supplement 3.** Characterization of SE–gene pairs for which nearest gene transcription remained unchanged, related to *Figure 4*.

in eRNA synthesis and a reduction of chromatin accessibility, with downregulation of eRNA synthesis preceding the decrease in accessibility (*Figure 5D*). This is illustrated for the SE near the *Klf4* gene, which contains three enhancers of this group (*Figure 5F*). For the second group, depletion of Oct4 led to a decrease in eRNA synthesis without changes in chromatin accessibility (*Figure 5E*). This is illustrated by the *Sox2* gene SE containing one enhancer of this group and *Mir290* SE (*Whyte et al., 2013*; *Figure 5G*, *Figure 5—figure supplement 1C*). The remaining Oct4-bound transcribed enhancers showed no changes in eRNA synthesis and chromatin accessibility upon Oct4 depletion (*Figure 5—figure supplement 1D*). We hereafter refer to the three groups of Oct4-bound transcribed enhancers as Oct4-sensitive, -insensitive, and -independent enhancers (*Figure 5C*). Taken together, these results indicate that reduced Oct4 binding leads to a delayed loss of chromatin accessibility at Oct4-sensitive enhancers compared to an immediate change of transcriptional activity.

## Oct4 cooperates with Sox2 to render enhancers accessible

To further characterize the differences between Oct4-sensitive and -insensitive enhancers, we analyzed publicly available ChIP-seq data. Oct4 and Sox2 colocalize in both enhancer groups, with 95% and 86% of Oct4-sensitive and -insensitive enhancers, respectively, overlapping with Sox2 peaks (*Figure 6—figure supplement 1A*). In metagene plots, Oct4-sensitive enhancers showed ~1.5-fold enrichment of Oct4 occupancy compared to Oct4-insensitive enhancers (*Figure 6A*), whereas Sox2, Nanog, Klf4, and Esrrb were only slightly enriched if at all (*Figure 6A*, *Figure 6—figure supplement 1B*). Oct4-sensitive enhancers also displayed higher levels of H3K27ac, whereas H3K4me1 showed similar levels (*Figure 6—figure supplement 1B*). Oct4-independent enhancers showed lower signals for pluripotency TFs and histone modifications, in line with the observed lower transcriptional activity (*Figure 6A*, *Figure 6—figure supplement 1B*). According to genomic region enrichment analysis (*McLean et al., 2010*), Oct4-sensitive enhancers were enriched for stem cell population maintenance (*Figure 6—figure supplement 1C*) and Oct4-insensitive enhancers for neural differentiation and development (*Figure 6—figure supplement 1D*). Enhancers of both types may target the same nearest active gene (*Figure 6—figure supplement 1E*).

To investigate whether a specific binding motif may be related to the enrichment of Oct4 occupancy in Oct4-sensitive enhancers, we performed motif analysis. Our results showed a strong enrichment for both Oct4 and the SoxOct composite motifs at Oct4-sensitive enhancers only (*Figure 6B*). These findings suggest that Oct4 influences chromatin accessibility preferentially when the SoxOct composite motif is present in DNA.

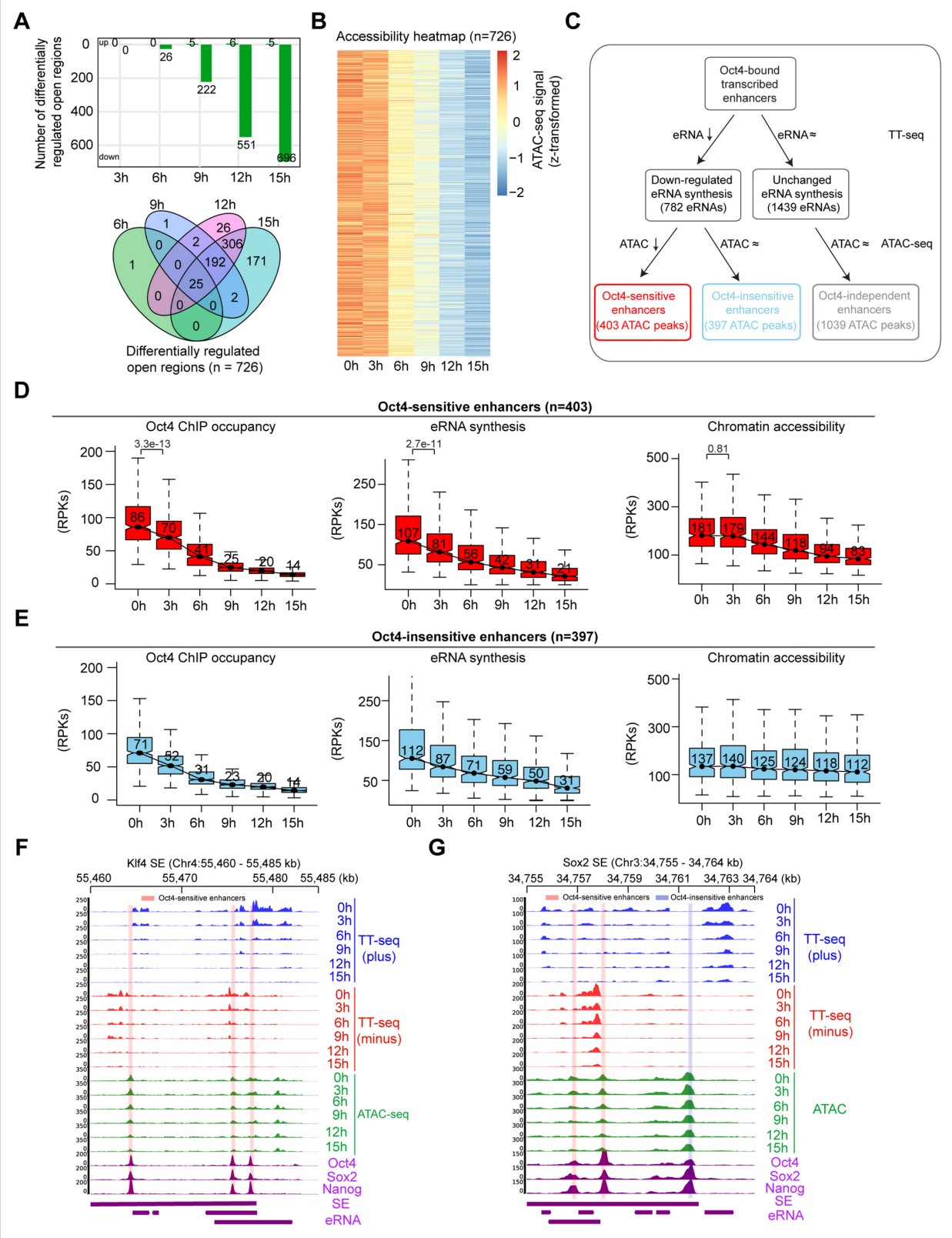

**Figure 5.** Reduced Oct4 binding does not immediately influence chromatin accessibility. (**A**) The number of differentially regulated chromatin open regions detected by DESeq2 for Oct4-bound transcribed enhancers (top) and Venn diagram showing overlapping of detected differentially regulated open regions (bottom) for each time point of doxycycline (DOX) treatment. (**B**) Heatmap visualizing the kinetics of chromatin accessibility changes at Oct4-occupied accessibility decreased sites (*n* = 726). (**C**) Diagram indicating classification of Oct4-sensitive, -insensitive, and -independent enhancers

*Figure 5 continued on next page*

*Figure 5 continued*

defined by change of eRNA synthesis and chromatin accessibility at Oct4-bound transcribed enhancers. (**D**) Boxplots indicating the changes of Oct4 occupancy, eRNA synthesis, and chromatin accessibility at Oct4-sensitive enhancers (*n* = 403). p values were calculated by Wilcoxon rank sum test. *y*-Axis represents read counts per kilobase (RPKs). Black bars represent the median values for each group. Lower and upper boxes are the first and third quartiles, respectively. The ends of the whiskers extend the box by 1.5 times the interquartile range. Outliers are omitted. (**E**) Boxplots indicating the changes of Oct4 occupancy, eRNA synthesis, and chromatin accessibility at Oct4-insensitive enhancers (*n* = 397). (**F**) Genome browser view for changes of eRNA synthesis and chromatin accessibility at *Klf4* superenhancer (SE) including three Oct4-sensitive enhancers. Tracks from top to bottom: TT-seq coverages of plus strand (blue), minus strand (red), and ATAC-seq coverages (green) at 0, 3, 6, 9, 12, and 15 hr; ChIP-seq coverages of Oct4, Sox2, and Nanog (purple) from ZHBTc4 mouse ES cell at 0 hr (*King and Klose, 2017*); SE annotation (*Whyte et al., 2013*); *Klf4* SE eRNAs annotated by TT-seq (purple arrow). Two biological replicates were merged for visualization. (**G**) Genome browser view for changes of eRNA synthesis and chromatin accessibility at *Sox2* SE including two Oct4-sensitive enhancers and one Oct4-insensitive enhancer. Tracks were ordered in the same way as in (**F**). Note that data from two biological replicates were generated for all assays and that the two replicates were merged for illustration.

The online version of this article includes the following figure supplement(s) for figure 5:

**Figure supplement 1.** Chromatin accessibility changes at Oct4-bound transcribed enhancers, related to *Figure 5*.

**Figure supplement 2.** Analysis of differential chromatin accessibility for Oct4-unbound transcribed enhancers, related to *Figure 5*.

## Sox2 remains transiently bound at Oct4-sensitive enhancers during Oct4 depletion

We then investigated whether depletion of Oct4 had an effect on Sox2 binding to enhancers. We performed ChIP-seq of Sox2 over the same time course (*Supplementary file 6*). At Oct4-sensitive enhancers, Sox2 remained bound from 0 to 6 hr and started to decrease after 9 hr of treatment (*Figure 6C, D*). At Oct4-insensitive, -independent, and -unbound transcribed enhancers, Sox2 occupancy was stable over the entire time course (*Figure 6C, D*, *Figure 6—figure supplement 2A, B*, *Figure 6—figure supplement 3*). Moreover, we analyzed published Oct4, Sox2, and Nanog ChIP-seq data at 0 and 24 hr after DOX treatment (*King and Klose, 2017*). Oct4-sensitive enhancers showed a strong loss of all three TFs at 24 hr, whereas Sox2 occupancy was unchanged and Nanog occupancy increased at 24 hr at Oct4-insensitive enhancers and -independent enhancers (*Figure 6—figure supplement 2C–E*). These findings are well illustrated at the exemplary genomic regions comprising SEs of *Klf4*, *Sox2*, and *Mir290* (*Figure 6E, F*, *Figure 6—figure supplement 2F*). Within these regions, we observed a decrease of chromatin accessibility and Sox2 occupancy at the Oct4-sensitive enhancers after 9 hr of DOX treatment (*Figure 6E, F*, *Figure 6—figure supplement 2F*), whereas the Oct4-insensitive enhancers remained accessible and occupied by Sox2 (*Figure 6F*, *Figure 6—figure supplement 2F*).

To gain insights into the change of chromatin accessibility and Sox2 binding at Oct4-bound sites without detected eRNA synthesis, we analyzed the 12,710 Oct4-bound nontranscribed enhancers (*Figure 3A, B*). PCA showed similar transitions for these enhancers over the time course as observed for Oct4-bound transcribed enhancers (*Figure 7—figure supplement 1A*). After 15 hr of DOX treatment, we detected 4985 enhancers with significantly reduced chromatin accessibility (*Figure 7—figure supplement 1B–D*, adjusted p value <0.01). We refer to them as Oct4-sensitive enhancers and the remaining ones as Oct4-insensitive enhancers (*Figure 7A*). For both enhancer groups, Oct4 occupancy decreased already after 3 hr of DOX treatment (*Figure 7B*), and for Oct4-sensitive enhancers the Oct4 occupancy decrease preceded the decrease in chromatin accessibility (*Figure 7B*, top).

We then investigated the effect of Oct4 depletion on Sox2 binding at Oct4-bound nontranscribed enhancers. Analysis of the Sox2-binding kinetics revealed a decrease of Sox2 binding at Oct4-sensitive enhancers after 9 hr of DOX treatment (*Figure 7C, D*). Oct4-insensitive enhancers showed only a slight decrease of Sox2 binding at 15 hr. Moreover, we analyzed published Oct4, Sox2, and Nanog ChIP-seq data at 0 and 24 hr after DOX treatment (*King and Klose, 2017*). Oct4-sensitive enhancers showed a strong loss of all three TFs at 24 hr, whereas at Oct4-insensitive enhancers Sox2 and Nanog occupancies were essentially unchanged (*Figure 7—figure supplement 1E*). This is consistent with our earlier observations at Oct4-bound transcribed enhancers (*Figure 6—figure supplement 2C–E*). Furthermore, metagene and motif analysis revealed an enrichment of Oct4 occupancy and the SoxOct composite motif at Oct4-sensitive enhancers (*Figure 7—figure supplement 1F, G*). Occupancy of other pluripotency factors and associated histone modifications also revealed a similar pattern as observed for Oct4-bound transcribed enhancers (*Figure 7—figure supplement 1F*). Taken together,

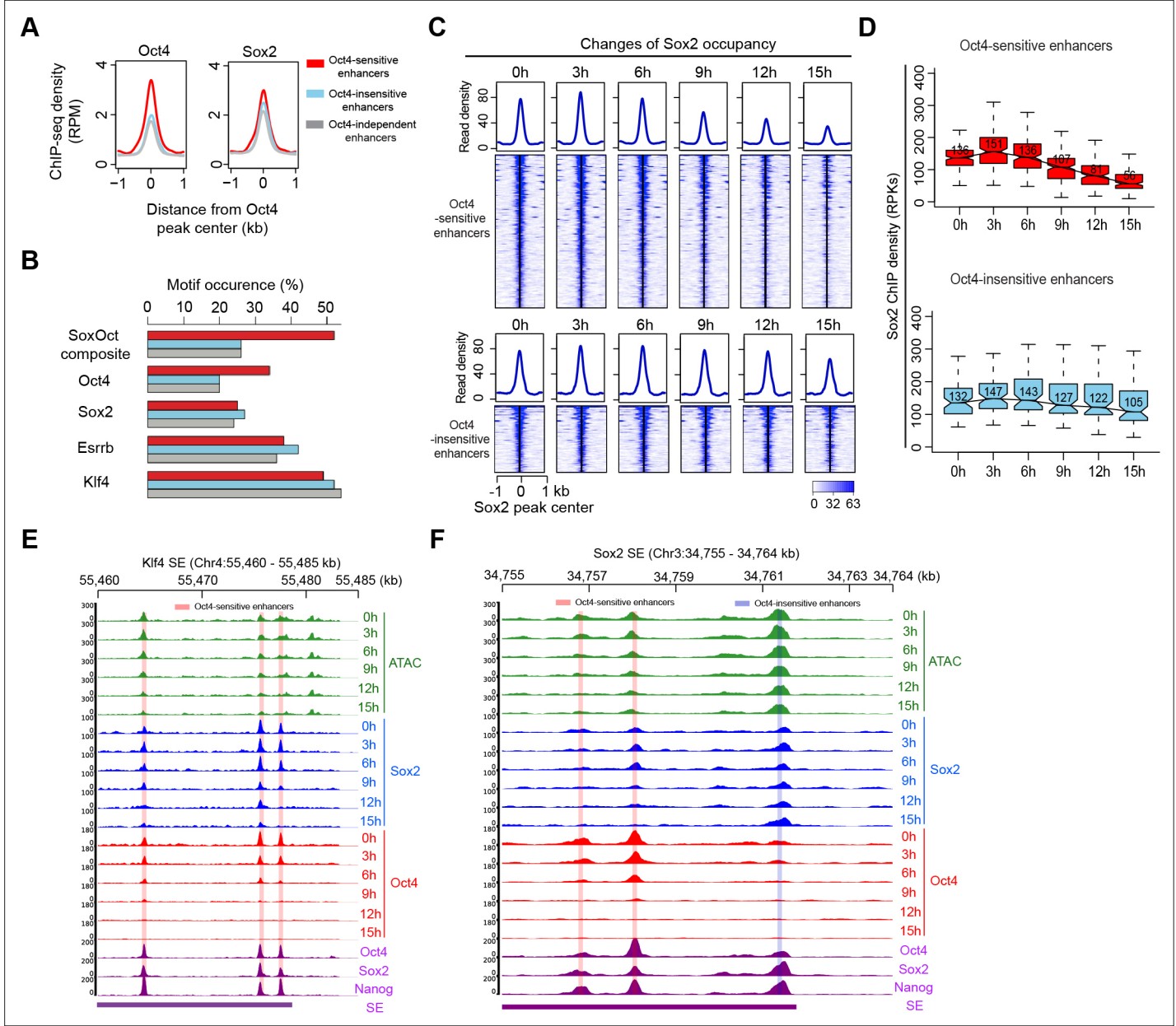

**Figure 6.** Sox2 remains bound transiently at Oct4-sensitive enhancers upon Oct4 depletion. (**A**) Metagenes analysis of Oct4 and Sox2 occupancy at Oct4-sensitive, -insensitive, and -independent enhancers at 0 hr, data were obtained from *King and Klose, 2017*. *y*-Axis depicts ChIP-seq coverage density in reads per million (RPM). (**B**) Percentage of motif occurrence at Oct4-sensitive, -insensitive, and -independent enhancers for SoxOct composite motif and Oct4, Sox2, Esrrb, and Klf4 motifs. (**C**) Heatmap showing changes of Sox2 occupancy at Oct4-sensitive and -insensitive enhancers over the entire time course of doxycycline (DOX) treatment. Normalized read densities are shown and peaks were ranked accordingly. (**D**) Same as (**C**), but using boxplots to depict quantification of Sox2 occupancy changes at Oct4-sensitive and -insensitive enhancers. *y*-Axis represents Sox2 ChIP-seq density in reads per kilobase (RPKs). Black bars represent the median values for each group. Lower and upper boxes are the first and third quartiles, respectively. The ends of the whiskers extend the box by 1.5 times the interquartile range. Outliers are omitted. (**E**) Genome browser view illustrating changes of chromatin accessibility, Sox2 and Oct4 occupancy at *Klf4* SE. Tracks from top to bottom: ATAC-seq coverages (green), ChIP-seq coverages for Sox2 (blue), and Oct4 (red) at 0, 3, 6, 9, 12, and 15 hr; ChIP-seq coverages for Oct4, Sox2, and Nanog (purple) from ZHBTc4 mouse ES cell at 0 hr (*King and Klose, 2017*); superenhancer (SE) annotation (*Whyte et al., 2013*). (**F**) Genome browser view illustrating changes of chromatin accessibility, Sox2 and Oct4 occupancy at *Sox2* SE. Tracks were visualized and ordered in the same way as in (**E**). Note that data from two biological replicates were generated for all assays and that the two replicates were merged for illustration.

The online version of this article includes the following figure supplement(s) for figure 6:

**Figure supplement 1.** Characterization of Oct4-sensitive, -insensitive, and -independent enhancers, related to *Figure 6*.

*Figure 6 continued on next page*

*Figure 6 continued*

**Figure supplement 2.** Change of Sox2 occupancy upon Oct4 depletion, related to *Figure 6*.

**Figure supplement 3.** Sox2 occupancy at Oct4-unbound transcribed enhancers, related to *Figure 6*.

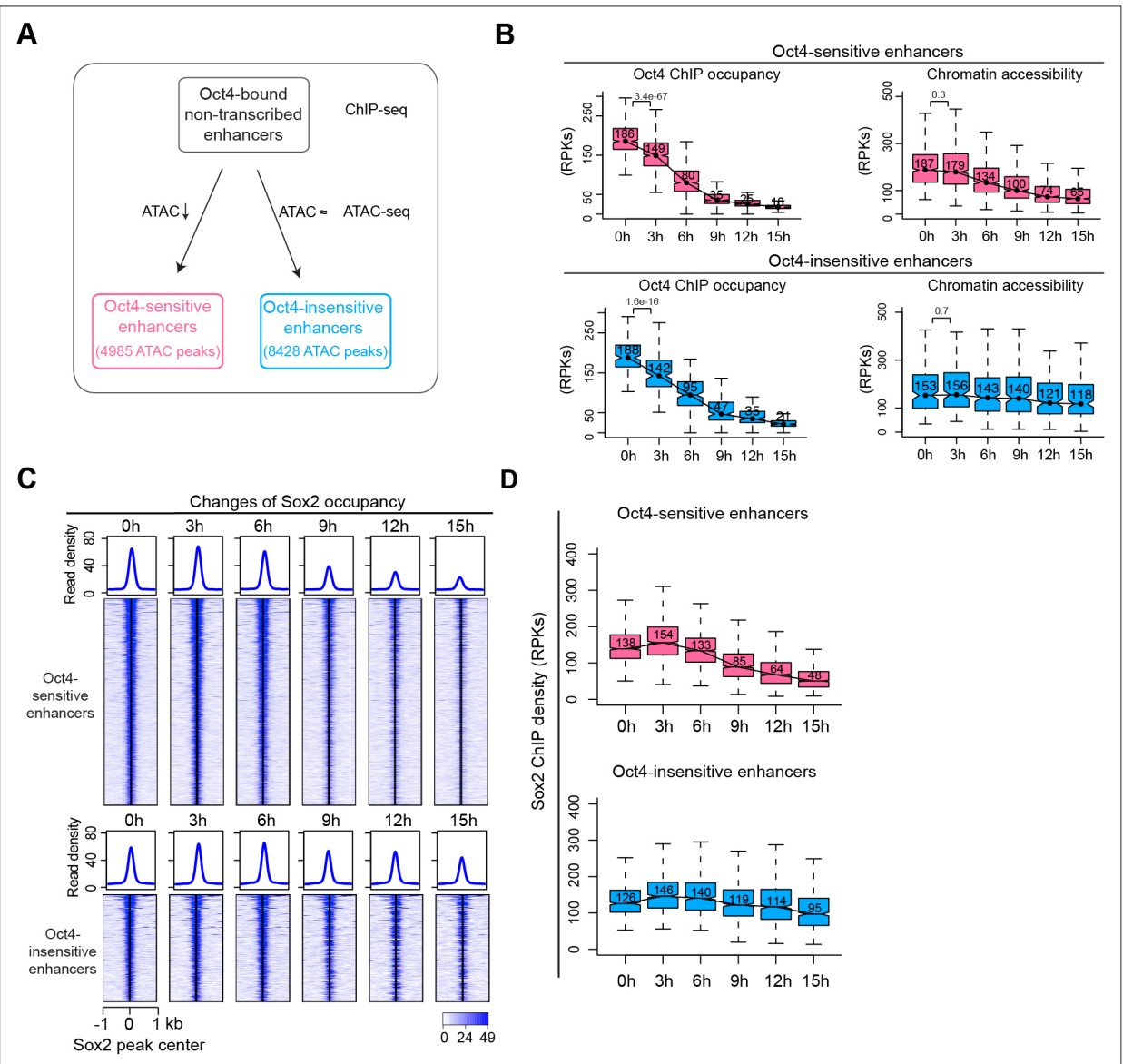

**Figure 7.** Sox2 remains bound transiently at Oct4-sensitive enhancers in the absence of eRNA synthesis. (**A**) Diagram indicating classification of Oct4-sensititive and -insensitive enhancers by changes of chromatin accessibility at Oct4-bound nontranscribed enhancers. (**B**) Boxplots illustrating changes of Oct4 occupancy and chromatin accessibility at Oct4-sensitive and -insensitive enhancers. p values were calculated by Wilcoxon rank sum test. *y*-Axis represents read counts per kilobases (RPKs). Black bars represent the median values for each group. Lower and upper boxes are the first and third quartiles, respectively. The ends of the whiskers extend the box by 1.5 times the interquartile range. Outliers are omitted. (**C**) Heatmap showing changes of Sox2 occupancy at Oct4-sensitive and -insensitive enhancers over the entire time course of doxycycline (DOX) treatment. Normalized read densities are shown and peaks were ranked accordingly. (**D**) Same as (**C**), but using boxplots to depict quantification of Sox2 occupancy changes at Oct4-sensitive and -insensitive enhancers. *y*-Axis represents Sox2 ChIP-seq density in reads per kilobase (RPKs). Note that data from two biological replicates were generated for all assays and that the two replicates were merged for illustration.

The online version of this article includes the following figure supplement(s) for figure 7:

**Figure supplement 1.** Chromatin accessibility changes at Oct4-bound nontranscribed enhancers, related to *Figure 7*.

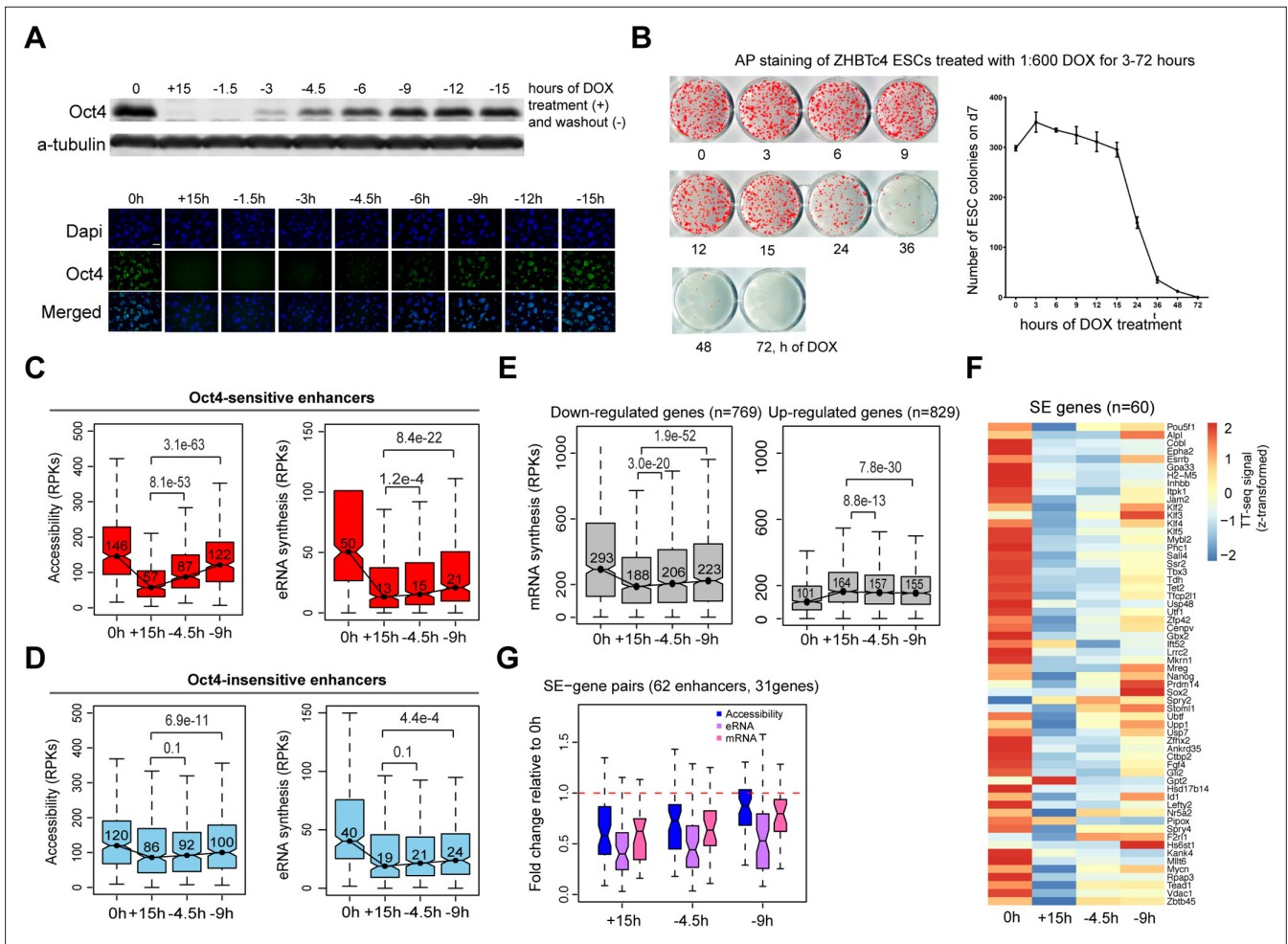

**Figure 8.** Partial recovery of Oct4 increases chromatin accessibility and enhancer transcription. (**A**) Western blot analysis of whole cell lysate samples after 0 and 15 hr of 1:600 diluted doxycycline (DOX) treatment and 1.5, 3, 4.5, 6, 9, 12, and 15 hr Oct4 recovery by DOX washout using Oct4 and a-tubulin antibodies (top). Oct4 protein levels decreased to 0.6% (+15 hr) and increased to 0.1% (−1.5 hr), 2.3% (−3 hr), 8.7% (−4.5 hr), 17.2% (−6 hr), 27.6% (−9 hr), 35.0% (−12 hr), and 37.0% (−15 hr) compared to 0 hr. Immunofluorescence of ZHBTc4 cells after 0 and 15 hr of 1:600 diluted DOX treatment and 1.5, 3, 4.5, 6, 9, 12, and 15 hr Oct4 recovery by DOX washout (bottom). Blue: DAPI, green: Oct4; scale = 200 µm. (**B**) AP staining of ZHBTc4 embryonic stem cells (ESCs) treated with 1:600 dox for 3–72 hr, 7 days after plating 1k cells on 12-well plate. (**C**) Boxplots indicating the changes in chromatin accessibility and eRNA synthesis at Oct4-sensitive enhancers during the time course of Oct4 depletion and recovery. p values were calculated by paired Wilcoxon rank sum test. y-Axis represents read counts per kilobase (RPKs). Black bars represent the median values for each group. Lower and upper boxes are the first and third quartiles, respectively. The ends of the whiskers extend the box by 1.5 times the interquartile range. Outliers are omitted. (**D**) As in (**C**), but for Oct4-insensitive enhancers. (**E**) Boxplots showing the changes of mRNA synthesis for previously identified downregulated genes (n = 769) and upregulated genes (n = 829) (**Figure 2B**) during Oct4 depletion and recovery. (**F**) Heatmap showing changes in mRNA synthesis for previously identified SE-controlled downregulated genes (n = 60, **Figure 2H**) during Oct4 depletion and recovery. (**G**) Boxplot showing fold changes in chromatin accessibility, eRNA and mRNA synthesis for previously identified SE–gene pairs (**Figure 4G**) during Oct4 recovery.

The online version of this article includes the following source data and figure supplement(s) for figure 8:

**Source data 1.** Source data for **Figure 8A**.

**Figure supplement 1.** Oct4 recovery rapidly restores pluripotency in ZHBTc4 mouse embryonic stem cells (ESCs), related to **Figure 8**.

**Figure supplement 1—source data 1.** Source data for **Figure 8—figure supplement 1A, B**.

these findings reveal that during Oct4 depletion Sox2 binding and chromatin accessibility change with similar kinetics at Oct4-bound enhancers.

## Partial recovery of Oct4 increases chromatin accessibility and enhancer transcription

To directly test the ability of Oct4 to control chromatin accessibility, we attempted Oct4 recovery experiments by washing out DOX. Using the original DOX concentration, Oct4 did not recover even after 24 hr of DOX washout, likely because of excessive DOX trapped in the cell membranes (*Figure 8—figure supplement 1A*). After optimization, we found that 1:600 dilution of conventional DOX concentration (1.67 ng/ml versus 1 μg/ml) delivers indistinguishable Oct4 depletion (*Figure 8—figure supplement 1B*) but allows a quick restoration after the washout (*Figure 8A*). We, therefore, chose 1:600 of DOX for the following recovery experiments. First, we tested if the pluripotent state can at all be restored after the absence of Oct4 (*Figure 8B*). The vast majority of ESCs could regain pluripotency after 15 hr of DOX treatment, fewer cells could recover after longer depletion of up to 48 hr, and none could recover after 72 hr. We showed that the few recovered ESCs after 48 hr of DOX treatment could be passed and maintain normal morphology (*Figure 8—figure supplement 1C*).

To study the earliest effects of Oct4 recovery, we collected time course data for recovery after 15 hr of DOX treatment (full Oct4 depletion) (*Supplementary files 7 and 8*). qPCR analysis confirmed the Oct4 depletion and restoration at the RNA level, and showed recovery of most pluripotency marker genes until 9 hr after the washout (*Figure 8—figure supplement 1D*), at which time Oct4 mRNA and protein levels were recovered to about 50% and 28%, respectively (*Figure 8—figure supplement 1D*, *Figure 8A*). TT- and ATAC-seq samples were collected after 4.5 and 9 hr of DOX washout; the biological replicates were highly reproducible (*Figure 8—figure supplement 1E*). At Oct4-sensitive enhancers, we found a strong increase of chromatin accessibility already after 4.5 hr when the Oct4 protein level was still only around 9% and the recovery of chromatin accessibility was mostly completed after 9 hr (*Figure 8C*). A weak but significant upregulation of eRNA synthesis occured at Oct4-sensitive enhancers (*Figure 8C*). At Oct4-insensitive enhancers, chromatin accessibility, and eRNA synthesis increased only slightly during Oct4 recovery (*Figure 8D*). At Oct4-independent and -unbound transcribed enhancers, chromatin accessibility and eRNA synthesis remained unchanged (*Figure 8—figure supplement 1F*).

We analyzed ATAC-seq data for 15 hr depletion and recovery samples using TOBIAS (*Bentsen et al., 2020*) that allows TF footprinting analysis for whole TF motif databases. In line with our previous motif analysis (*Figure 6B*), at 15 hr of DOX treatment, SoxOct footprints were the most significantly depleted in ATAC-seq signal, followed by Oct alone footprints (*Figure 8—figure supplement 1G*). This confirms that the changes in chromatin accessibility are driven by the loss of Oct4 and Sox2 binding. Consistently, SoxOct and Oct footprints were the most significantly enriched after 4.5 hr of recovery (*Figure 8—figure supplement 1H*). Interestingly, later into the recovery process – after 9 hr of washout, open chromatin was enriched with not just SoxOct and Oct footprints, but also Klf footprints (*Figure 8—figure supplement 1I*), inferring the role of Klf TFs in re-establishing of pluripotency after Oct4 depletion.

Finally, we investigated the effects of partial Oct4 recovery on mRNA synthesis for genes that were differentially expressed upon Oct4 depletion (*Figure 2A, B*). Whereas mRNA synthesis of previously upregulated genes decreased only slightly, previously downregulated genes showed an increase of mRNA synthesis during Oct4 recovery (*Figure 8E*). In particular, for most of the SE-controlled downregulated genes (*Figures 2H and 4G*), we observed an increase in mRNA synthesis (*Figure 8F*), accompanied by an increase in chromatin accessibility and eRNA synthesis at their paired enhancers (*Figure 8G*). Taken together, these results indicate that partial recovery of Oct4 rapidly increases chromatin accessibility, followed by rebound of gene transcription activity, with the recovery of associated enhancer transcription being weaker.

## Discussion

Here, we used Oct4 depletion from mouse ESCs and several genomic approaches to analyze the role of Oct4 in the maintenance of pluripotency over a high-resolution time course. We used a Tet-off system that allows gradual Oct4 depletion (*Niwa et al., 2000*), within 15 hr. The gradual depletion

provided insights into Oct4's concentration-dependent role in gene expression regulation. Our data showed that loss of Oct4 from enhancers goes along with a decrease in mRNA synthesis from Oct4 target genes crucial for maintaining pluripotency.

During Oct4 depletion, enhancer and target gene transcription were very sensitive to even small reductions of Oct4 binding, whereas chromatin accessibility was either unaffected (Oct4-insensitive enhancers) or affected later when Oct4 levels decreased considerably (Oct4-sensitive enhancers). During partial recovery of Oct4, chromatin accessibility and enhancer transcription increased at Oct4-sensitive enhancers, but eRNA synthesis could not be fully reactivated, although chromatin accessibility was largely restored. Together, this indicates that normal ESC levels of Oct4 are required for enhancer transcription whereas low levels of Oct4 are sufficient to retain chromatin accessibility. This is also consistent with a recent study, in which the authors used an Oct4-low expressed protein depletion system in ESCs and observed that loss of chromatin accessibility was quasi-synchronized with Oct4 loss (*Friman et al., 2019*). As shown previously, ESCs with reduced Oct4 expression maintained a robust naive pluripotent state, but only wild-type Oct4 levels could enable effective differentiation to all embryonic lineages (*Karwacki-Neisius et al., 2013*; *Radzisheuskaya et al., 2013*). It would be interesting to investigate if Oct4-induced eRNAs play a role in the stabilization of the pluripotency network, or in the capacity of the Oct4-high, but not Oct4-low, ESCs to differentiate (*Chang et al., 2019*; *Fallatah et al., 2021*; *Zhao et al., 2019*).

Oct4 is thought to activate pluripotency genes by forming phase-separated condensates at SEs (*Boija et al., 2018*). This model is consistent with our observation that high levels of Oct4 are required to maintain and reactivate transcription in recovery experiments. A recent study suggests that TFs may bind DNA to form condensates and once the binding is saturated at high concentrations, the extra molecules can be absorbed into the condensates directly or via one-dimensional diffusion and therefore direct DNA binding is not required (*Morin et al., 2022*). Based on such a model, it could be possible that low levels of Oct4 are sufficient to maintain its interaction pattern with DNA, whereas at high levels the saturated Oct4 molecules can be incorporated into the condensates directly and contribute to transcription activation.

Oct4 cooperates with Sox2 to regulate the vast majority of its targets in pluripotent cells (*Chen et al., 2014*; *Chronis et al., 2017*; *Malik et al., 2019*; *Reményi et al., 2003*). While Sox2 protein level remained unchanged throughout the 15 hr of DOX treatment, Sox2 was depleted from chromatin starting from 9-hr DOX treatment. Accordingly, at enhancers dependent on Oct4 for their accessibility (Oct4-sensitive enhancers) Sox2 remained transiently bound during the first 6 hr of DOX treatment and only decreased when chromatin accessibility also decreased. This can be explained by the proposed model that low levels of Oct4 are sufficient to retain chromatin accessible, thus leading to a loss of Sox2 binding only when Oct4 levels decreased to the point where chromatin accessibility can no longer be maintained. This befits the pioneer role of Oct4 in creating an open chromatin environment to support binding of other pluripotency factors (*Friman et al., 2019*; *King and Klose, 2017*). In addition, reduced levels of Oct4 may cooperate with Sox2 to temporarily retain enhancers accessible. Furthermore, we found enhanced binding of Sox2 at Oct4-bound enhancers after 3 hr of DOX treatment. This suggests that Oct4 may inhibit Sox2 binding at normal levels in ESCs, and agrees with a study that has shown that overexpression of Oct4 decreases Sox2 binding (*Biddle et al., 2019*). This is similar to a recent study in which enhanced binding of Nanog was observed after complete degradation of Oct4 protein (*Bates et al., 2021*).

Our TF footprinting analysis detected a significant enrichment for Klf footprints at the later stages of Oct4 recovery. Klf TFs are at the core of pluripotency network and are crucial for induction of pluripotency (*Nakagawa et al., 2008*). Klf4 and Klf5 directly interact with Oct4 and Sox2 to promote reprogramming to iPSCs and maintenance of pluripotency in ESCs (*Han et al., 2022*; *Wei et al., 2009*). Oct4 cooperates with chromatin remodeler Brg1 to loosen the chromatin allowing Klf4 binding that facilitates reprogramming to pluripotency (*Chen et al., 2020*). Taken together, this implies that Oct4 creates an open chromatin environment to prepare for binding of Klf TFs to support further maturation of the pluripotency network during Oct4 recovery.

In a recent study, Oct4 protein was fully depleted within 2 hr using an auxin-inducible degron system (*Bates et al., 2021*). Using RNA-seq, the authors showed that full depletion of Oct4 leads to rapid downregulation of key pluripotency factors, consistent with our results (*Figure 2H*). We used TT-seq, which is more sensitive than RNA-seq and directly detects changes in gene transcription

upon perturbation (*Michel et al., 2017*; *Schwalb et al., 2016*). Thus, although Oct4 was depleted in different ways, both studies support the same conclusion that loss of Oct4 results in rapid downregulation of pluripotency-associated genes.

Analysis of the TT-seq data revealed that the used ZHBTc4 cells were contaminated with mycoplasma (Materials and methods). However, this does not alter the conclusions described in this manuscript. First, the ZHBTc4 cells exhibit normal growth patterns and morphology (*Figure 1—figure supplement 1A*). Second, our experiments and analyses were all done in relation to the untreated ZHBTc4 cells. Third, as discussed, our results confirm previously published observations with mycoplasma-negative mESCs (*Friman et al., 2019*; *King and Klose, 2017*; *Bates et al., 2021*).

In summary, we investigated the role of Oct4 in regulating chromatin accessibility and transcriptional activity in pluripotent cells using high-resolution time course data for Oct4 depletion and subsequent recovery. We discovered that Oct4 has differential concentration-dependent activities in regulating enhancer transcription and chromatin accessibility in pluripotent stem cells. Normal ESC concentrations of Oct4 are required to maintain the transcription of pluripotency enhancers. In contrast, Oct4 regulates chromatin accessibility very rapidly even at low concentrations, capable of increasing chromatin accessibility within hours of induction.

# Materials and methods

**Key resources table**

| Reagent type (species) or resource | Designation | Source or reference | Identifiers | Additional information |
|---|---|---|---|---|
| Cell line (*Mus musculus*) | ESC_ZHBTc4: *Pou5f1*[BSD/Zeo]; Tg(CAG-tTA); Tg(TetO-*Pou5f1*) | *Niwa et al., 2000* | RRID:CVCL_C715 | Oct4 depletion cell line |
| Antibody | Mouse monoclonal against Oct-3/4 | Santa Cruz | Cat# sc-5279, RRID:AB_628051 | Western (1:2500) |
| Antibody | Goat polyclonal against Sox2 | Santa Cruz | Cat# sc17320, RRID:AB_2286684 | Western (1:1000) |
| Antibody | Rabbit polyclonal against Nanog | Bethyl Laboratories | Cat# A300-397A, RRID:AB_386108 | Western (1:5000) |
| Antibody | Rabbit polyclonal against Histone 3 | Abcam | Cat# ab1791, RRID:AB_302613 | Western (1:20000) |
| Antibody | Mouse monoclonal against anti-alpha-Tubulin | Sigma-Aldrich | Cat# T6199, RRID:AB_477583 | Western (1:5000) |
| Antibody | Mouse monoclonal against Oct-3/4 | Santa Cruz | Cat# sc-5279, RRID:AB_628051 | IF (1:1000) |
| Antibody | Goat polyclonal against Sox2 | Santa Cruz | Cat# sc17320, RRID:AB_2286684 | IF (1:500) |
| Antibody | Goat polyclonal against Sox2 | Neuromics | Cat# GT15098, RRID:AB_2195800 | ChIP (1.5 ug/25 ug chromatin) |
| Antibody | Goat polyclonal against Oct-3/4 | R&D | Cat# AF1759, RRID:AB_354975 | ChIP (4 ug/25 ug chromatin) |
| Commercial assay or kit | Ovation Universal RNA-seq System | NuGEN | Cat# 0343-32 | |
| Commercial assay or kit | NEBNext Ultra II DNA Library Prep Kit | NEB | Cat# E7370L | |
| Commercial assay or kit | Illumina Tagment DNA Enzyme and Buffer large kit | Illumina | Cat# 20034198 | |
| Commercial assay or kit | Nextera Tn5 enzyme | Illumina | Cat# 20034198 | |
| Chemical compound, drug | CHIR99021 | Cayman Chemicals | 13,122 | |
| Chemical compound, drug | PD0325901 | Biomol | 103034-25 | |
| Chemical compound, drug | Doxycycline | Sigma-Aldrich | Cat# D9891-1G | |
| Chemical compound, drug | 4-Thiouridine | Carbosynth | Cat# NT06186 | |
| Chemical compound, drug | Formaldehyde 16% concentrate stock methanol-free | Thermo Fisher Scientific | Cat# 28,908 | |

*Continued on next page*

*Continued*

| Reagent type (species) or resource | Designation | Source or reference | Identifiers | Additional information |
|---|---|---|---|---|
| Software, algorithm | STAR (2.5.3) | *Dobin et al., 2013* | RRID:SCR_015899 | |
| Software, algorithm | Samtools (1.6) | *Li et al., 2009* | RRID: SCR_002105 | |
| Software, algorithm | HTSeq (0.9.1) | *Anders et al., 2015* | RRID:SCR_005514 | |
| Software, algorithm | GenoSTAN (2.18.0) | *Zacher et al., 2017* | https://www.bioconductor.org/packages/release/bioc/html/STAN.html | |
| Software, algorithm | tilingArray (1.68.0) | *Huber et al., 2006* | https://www.bioconductor.org/packages/release/bioc/html/tilingArray.html | |
| Software, algorithm | pyGenomeTracks (3.5.1) | *Lopez-Delisle et al., 2021* | RRID:SCR_016366 | |
| Software, algorithm | Cutadapt (1.16) | *Martin, 2011* | RRID:SCR_011841 | |
| Software, algorithm | MACS2 (2.1.1.20160309) | *Zhang et al., 2008* | RRID:SCR_013291 | |
| Software, algorithm | Bowtie 2 (2.3.4.1) | *Langmead and Salzberg, 2012* | RRID:SCR_005476 | |
| Software, algorithm | DESeq2 (1.26.0) | *Love et al., 2014* | RRID:SCR_015687 | |
| Software, algorithm | DAVID Bioinformatcis Resources (6.8) | *Huang et al., 2009* | RRID:SCR_001881 | |
| Software, algorithm | GREAT (4.04) | *McLean et al., 2010* | RRID:SCR_005807 | |
| Software, algorithm | FIMO (5.0.4) | *Grant et al., 2011* | RRID:SCR_001783 | |
| Software, algorithm | UCSD LiftOver | *Hinrichs et al., 2006* | RRID:SCR_018160 | |
| Software, algorithm | TOBIAS | *Bentsen et al., 2020* | https://github.com/loosolab/TOBIAS | |

## Employed cell lines and DOX treatment

ZHBTc4 mESCs harbouring a DOX repressible *Pou5f1* transgene (*Niwa et al., 2000*) were propagated on gelatin-coated plates in equal parts DMEM-F12 (Life Technology, 21331-020) and neural-basal (Life Technology, 21103-049) medium supplemented with 2% fetal bovine serum (Sigma-Aldrich, G1393-100 ml), 2% knockout serum replacement medium (Gibco, 10828-028), 0.04 µg/ml leukemia inhibitory factor (prepared in-house), penicillin/streptomycin (Sigma-Aldrich, P4333-100ml), 0.1 mM β-mercaptoethanol (Gibco, 31350-010), 0.5× B27 supplement (Life Technology, 12587-010), 0.5× N2 supplement (Gibco, AM9759), 3 µM CHIR99021 (Cayman Chemicals, 13122), and 1 µM PD0325901 (Biomol, 103034-25). Cells were passaged using Accutase (Sigma-Aldrich, 6964–100 ml) DOX was used at 1 µg/ml for the depletion experiments or 1.67 ng/ml for the recovery experiments (Sigma-Aldrich D9891-1G). The ZHBTc4 mESCs were authenticated through selection using neomycin as well as via treatment with DOX to check for responsiveness of the Oct4 TET-OFF transgene. Subjecting TT-seq data of the untreated samples to OpenContami (https://openlooper.hgc.jp/opencontami/), we found that ~0.3% of the reads mapped to mycoplasma, revealing that the ZHBTc4 cells used were mycoplasma positive.

## Sample preparation and western blotting

ZHBTc4 cells were washed with phosphate-buffered saline (PBS) and harvested using Accutase (Sigma-Aldrich, 6964–100 ml) at the given time points of loss-of-Oct4. Cells were centrifuged for 5 min at 1400 rpm, the supernatant was aspirated and the cell pellet resuspended as a single-cell suspension in cell culture medium. For whole cell lysate, cell pellets were weighed. Cell pellets were resuspended in 4× LDS buffer (prepared in-house) based on weight. Equal volume for each sample was loaded on to SDS–PAGE gels. For chromatin samples, cell number was determined using a counting chamber. 2 × $10^7$ cells were crosslinked with a final concentration of 1% formaldehyde AppliChem, A0877,0250 for 8 min and quenched for 5 min with 125 mM glycine (Sigma-Aldrich, G8898-1KG). Crosslinked cells were centrifuged for 5 min at 1350 × *g* and washed twice with 1 ml of cold PBS. Cells were either stored at −80°C or directly processed for chromatin extraction. For whole cell lysate samples, equal volume for each sample was loaded on to sodium dodecyl sulfate polyacrylamide gel electrophoresis (SDS–PAGE) gels. Chromatin samples were loaded equally based on DNA concentrations. Blots were

probed for Oct4 (Santa Cruz, sc-5279), Sox2 (Santa Cruz, sc17320), Nanog (Bethyl Laboratories, A300-397A), Histone 3 (H3) for chromatin samples (Abcam ab1791), Tubulin for whole cell lysate samples (Sigma-Aldrich, T6199) overnight at 4°C. The next day, blots were washed and probed with secondary antibodies anti-mouse-HRP (Jackson Labs, 115-035-044), anti-goat-HRP (R&D systems, HAF019), and anti-rabbit-HRP (GE Healthcare, NA934) at room temperature for 2 hr. Blots were exposed to film using ECL (GE Healthcare, RPN2232).

## Immunofluorescence

ZHBTc4 cells were treated for 0 and 24 hr. Next, cells were crosslinked with 4% paraformaldehyde (Sigma-Aldrich, D6148-500G) for 30 min. Formaldehyde was quenched with 50 mM glycine (Sigma-Aldrich, G8898-1KG) for 15 min. Cells were stained for Oct4 (Santa Cruz sc-5279) or Sox2 (Santa Cruz, sc17320) overnight at 4°C. The following day samples were incubated with anti-goat-alexa-488 (Thermoscientific, A11078) or anti-goat-alexa-568 (Thermoscientific, A11061) and Hoechst (Sigma-Aldrich, H6024).

## TT-seq

TT-seq was performed as described (*Schwalb et al., 2016*) with minor alterations. In brief, two biological replicates at the aforementioned timepoints were produced for TT-seq. $1 \times 10^8$ cells were labeled with 500 µM (4sU Carbosynth, 13957-31-8) for 5 min. Cells were harvested and lysed using TRIzol (Ambion, 1559018) and stored at −80°C. Prior to RNA isolation, RNA spike-ins were added at 5 ng per $1 \times 10^8$ cells. Details regarding the used spike-in sequences and generation of the spike-in mix can be found in *Wachutka et al., 2019*. Total RNA was isolated using TRIzol (Ambion, 1559018) according to the manufacture's instructions, and subsequentially fragmented to 1500–5000 bp using Covaris S220 Ultrasonicator. Nascent RNA was purified as described (*Schwalb et al., 2016*) with minor modifications. Following purification using streptavidin pulldown, the collected RNA was purified using RNeasy micro Kit (Qiagen, 74004), as well as DNase treatment (Qiagen, 79254). Sequencing libraries were produced using NuGen Ovation Universal RNA-seq System (Nugen, 0343). Size-selected libraries were analyzed on a Fragment Analyzer before sequencing on an Illumina NEXTseq 550.

## ATAC-seq

ATAC-seq was performed as described (*Buenrostro et al., 2013*) with a few alterations. ZHBTc4 cells were harvested using Accutase (Sigma-Aldrich, 6964–100 ml) at 0, 3, 6, 9, 12, and 15 hr. Nuclear isolation of $5 \times 10^4$ cells was followed by treatment with Nextera Tn5 enzyme (Illumina, 20034198) for 45 min at 37°C. PCR amplification of the samples was performed using Nextera primers 1 and 2 and NEBNext High fidelity master mix (NEB, M0541S) for 12 cycles as determined by KAPA Real-Time Library Amplification Kit (Peqlab, KK2701). Libraries were purified over Macherey-Nagel PCR spin column (Macherey-Nagel, 740609.50S) and AMPure XP beads (Beckman Coulter, A63881) in a 1:1.8 ratio. Sequencing of libraries was performed on an Illumina NEXTseq 550.

## Sox2 ChIP-seq

Crosslinked pellets (as described in sample preparation for western blot) were thawed on ice. Protease inhibitor (Roche, 4693124001) was added to all buffers. Pellets were resuspended in lysis buffer 1 (50 mM HEPES(4-(2-hydroxyethyl)-1-piperazineethanesulfonic acid)–KOH pH 7.5, 140 mM NaCl, 1 mM EDTA pH 8, 10% glycerol, 0.5% IGEPAL CA630, 0.25% Triton X-100) and lysed for 30 min on ice. Samples were pelleted and washed with lysis buffer 2 (10 mM Tris–HCl pH 8, 200 mM NaCl, 1 mM EDTA(Ethylene diamine tetraacetic acid), 0.5 M EGTA(Ethylene glycol tetraacetic acid)) for 10 min on a roller bank at 4°C. Samples were pelleted and resuspended in SDS sonication buffer (10 mM Tris–HCl, 1 mM EDTA, 0.5% SDS), incubated on ice for 10 min and transferred to TPX sonication tubes (Diagenode, C30010009). Chromatin was sonicated in Diagenode Bioruptor 4 × 15 min at 30 s ON and 30 s OFF, high setting in a cooled water bath. Sheared chromatin was centrifuged for 10 min at 15,000 rpm at 4°C. 25 µl of the sample was de-crosslinked overnight at 65°C and the distribution of size was checked on 1.4% agarose gel. 3 µg of Sox2 antibody (Neuromics, GT15098) was coupled to Dynabeads protein G (Thermo Fisher Scientific, 10,009D) for 2 hr at 4°C for each sample. 50 µg of chromatin was used for each immunoprecipitation (IP). Chromatin was diluted using ChIP dilution buffer (10 mM Tris–HCl pH 8, 125 mM NaCl, 0.125% sodium deoxycholate, 1.25% Triton

X-100). Antibody–chromatin mix was incubated overnight at 4°C rotating end-over-end. Samples were washed with low salt buffer (20 mM Tris–HCl pH 8, 150 mM NaCl, 2 mM EDTA, 0.1% SDS, 1% Triton X-100), twice using high salt buffer (20 mM Tris–HCl pH 8, 500 mM NaCl, 0.1% SDS, 1% Triton X-100), twice using RIPA washing buffer (50 mM HEPES–KOH pH7.6, 250 mM LiCl, 1 mM EDTA, 1% IGEPAL CA630, 0.7% sodium deoxycholate) and once with TE buffer containing 50 mM NaCl. Bound chromatin was eluted using 105 µl prewarmed elution buffer (10 mM Tris–HCl pH 8, 5 mM EDTA, 300 mM NaCl, 0.5% SDS) for 15 min at 65°C. RNase A Invitrogen, 1,004D was added and the samples were incubated overnight at 65°C. The next day, samples were treated with Proteinase K (AppliChem, A4392,0010) for 2 hr at 55°C. Samples were purified using the Macherey-Nagel PCR spin column (Macherey-Nagel, 740609.50S). DNA quantity was done using Qubit 3.0 (Life Technology, Q33126). 25 ng of DNA was used to prepare sequencing libraries using NEBNext Ultra DNA Library Prep Kit (NEB, E7370L) according to the manufacture's manual. Purity and size distribution were analyzed on Fragment Analyzer. Libraries were sequenced on a HiSeq 1,500 (Illumina).

## Oct4 ChIP-seq
Crosslinked pellets (as described in sample preparation for western blot) were thawed on ice. Protease inhibitor (Roche, 4693124001) was added to all buffers. A pellet coming from $3 \times 10^7$ cells was resuspended in Farnham Lysis buffer (5 mM Pipes pH 8, 85 mM KCl, 0.5% NP-40) and lysed for 10 min on ice. Samples were pelleted for 5 min at $1700 \times g$ at 4°C. Samples were washed with PBS and pelleted for 5 min at 1700 $g$ at 4°C. Samples were resuspended in 1 ml of SDS sonication buffer (10 mM Tris–HCl 7.5 pH, 1 mM EDTA, 0.4% SDS), incubated on ice for 10 min and transfer to AFA milliTube. Sonication was performed with a S220 Focused-ultrasonicator (Covaris) with the following parameters: duty cycle 5%, peak incident power 140 W, cycle per burst 200, processing time 840 s, degassing mode continuous, water run level 8. 25 µl of the sample was de-crosslinked overnight at 65°C and the distribution of size was checked on 1.4% agarose gel. 40 µg of Oct4 antibody (R&D, AF1759) was coupled to Dynabeads protein G (Thermo Fisher Scientific, 10,009D) for 2 hr at room temperature for each sample. 100 µg of chromatin was used for each IP. 100 ng of *Drosophila S2* sheared crosslinked chromatin (Covaris S200 parameters: duty cycle 5%, peak incident power 140 W, cycle per burst 200, processing time 1800 s, degassing mode continuous, water run level 8) were added to 100 µg of chromatin as spike-ins control. Chromatin was diluted in IP buffer (50 mM HEPES pH 7.9, 150 mM NaCl, 1 mM EDTA, 1% Triton X-100, 0.1% sodium deoxycholate) to obtain a 0.1% final concentration of SDS. 1% of diluted chromatin was kept as input at 4°C. Antibody–chromatin mix was incubated overnight at 4°C rotating end-over-end. Samples were washed five times with IP wash buffer (100 mM Tris–HCl pH 7.5, 500 mM LiCl, 1% NP-40, 1% sodium deoxycholate) and one time with TE buffer (10 mM Tris–HCl pH 8, 1 mM EDTA). Immuno-bound chromatin was eluted at 70°C for 10 min with elution buffer (0.1 M NaHCO$_3$, 1% SDS) and de-crosslinked overnight at 65°C. After RNAse A treatment at 37°C for 1.5 hr and proteinase K treatment at 45°C for 2 hr, DNA was extracted with one volume phenol:chloroform:isoamyl alcohol 25:24:1 (Sigma-Aldrich, P2069) and precipitated for 30 min at −80°C with 200 mM NaCl and 100% ethanol. Pellet was washed with 70% ethanol and resuspended in TE buffer. DNA quality and size distribution were checked on Fragment Analyzer. 3 ng of DNA was used for library preparation according to NEBNext Ultra II DNA Library Prep Kit (NEB, E7645S). Purity and size distribution were analyzed on Fragment Analyzer. Size-selected libraries were sequenced on Illumina NEXTseq 550.

## TT-seq data preprocessing
Paired-end 42 bp reads were mapped to the mouse genome assembly mm10 using STAR 2.5.3 (*Dobin et al., 2013*) with the following parameters: outFilterMismatchNmax 2, outFilterMultimapScoreRange 0, and alignIntronMax 500,000. SAMtools (*Li et al., 2009*) was then used to remove alignments with MAPQ smaller than 7 (-q 7) and only proper pairs (-f 2) were selected. HTSeq-count (*Anders et al., 2015*) was used to calculate fragment counts for different features. Further data processing was carried out using the R/Bioconductor environment.

## TU annotation and classification
Annotation of TU was performed as described (*Schwalb et al., 2016*) with minor modifications. Briefly, the whole genome was segmented into 200 bp consecutive bins and the midpoint of TT-seq fragments

was then used to calculate the coverage for each bin for each sample. A pseudo-count was added to each bin to avoid noisy signals. In order to create a unified annotation independent of a specific time point, all TT-seq samples were combined. The R/Bioconductor package GenoSTAN (*Zacher et al., 2017*) was then used to learn a two-state hidden Markov model with a PoissonLog-Normal emission distribution in order to segment the genome into 'transcribed' and 'untranscribed' states. Transcribed regions overlapping at least 20% of their length with GENCODE annotated protein-coding gene or lincRNA and overlapping with an annotated exon were classified as mRNA/lincRNA and the rest was defined as ncRNA. Transcribed regions mapping to exons of the same protein-coding gene or lincRNA were combined to create a consecutive TU. In order to avoid spurious predictions, ATAC-seq data were used to call open chromatin regions (see below) and TUs without their promoter (±1 kb of transcription start site, TSS) overlapping with an opening chromatin region were removed (*Figure 1—figure supplement 1D*). A minimal expression threshold was optimized based on expression difference between TUs with or without their promoter overlapping an opening chromatin region. This resulted in 26822 TUs originating from an open chromatin region with a minimal RPK of 26.5 (*Figure 1—figure supplement 1E*). In order to overcome low expression or mappability issues, ncRNAs that are only 200 bp (1bin) apart were merged. Subsequently, TU start and end sites were refined to single nucleotide precision by finding borders of abrupt coverage increase or decrease between two consecutive segments in the four 200 bp bins located around the initially assigned start and stop sites via fitting a piecewise constant curve to the TT-seq coverage profiles for both replicates using the segmentation method from the R/Bioconductor package (*Huber et al., 2006*) ncRNAs were then classified into the following four categories according to their respective location relative to protein-coding genes: upstream antisense RNA (uaRNA), convergent RNA (conRNA), antisense RNA (asRNA), and intergenic RNA (incRNA) (*Figure 1C*). ncRNAs located on the opposite strand of an mRNA were classified as asRNA if the TSS was located >1 kbp downstream of the sense TSS, as uaRNA if the TSS was located <1 kbp upstream of the sense TSS, and as conRNA if the TSS was located <1 kbp downstream of the sense TSS. The remaining ncRNAs were classified as incRNA.

To annotate putative eRNAs we selected asRNAs and incRNAs. Since highly synthesized mRNAs can give rise to spurious and uncontinuous downstream transcription signal, we restricted the analysis to a subset of asRNA and incRNA that are located 1 kb far away from promoter related RNAs including mRNA, uaRNA, conRNA, and defined them as putative eRNA. We then merged them if the putative eRNAs fell within 1 kb of each other. eRNAs within 1 kb of Oct4-occupied opening regions were defined as Oct4-regulated eRNAs and the corresponding Oct4 peaks were classified as Oct4-bound transcribed enhancers. Genome browser views for coverages and annotations were plotted by software pyGenomeTracks (*Lopez-Delisle et al., 2021*).

## Differential gene expression analysis

R/Bioconductor package DESeq2 (*Love et al., 2014*) was used to call differentially expressed mRNAs and eRNAs applying DESeq2's default size factor normalization. For both cases DESeq2 size factors were calcualted using counts for protein-coding genes. An adjusted p value of 0.01 was used to identify significantly changed mRNAs or eRNAs by comparing each time point to the 0-hr measurement.

## Principal component analysis

For each replicate, size factor normalized feature counts were obtained and DESeq2's default variance stabilizing transformation was applied. The DESeq2 plotPCA function was used to generate PCA plots with the default parameters.

## *k*-Means clustering

Size factor normalized feature counts were aggregated for two biological replicates for each time point and then the data matrix was subjected to *z*-score transformation before clustering. *k*-Means clustering was then performed using the kmeans function in R.

## ATAC-seq data processing

Paired-end 76 bp reads were obtained for each of the samples and Nextera Transposase adapter sequence was removed using (*Martin, 2011*). Bowtie2 (*Langmead and Salzberg, 2012*) was used to align paired-end reads to the mouse genome assembly mm10 with the '`--local`' and

'`--no-discordant`' options. SAMtools (*Li et al., 2009*) was then used to remove alignments with MAPQ smaller than 7 (-q 7) and only proper pairs (-f 2) were selected. Reads mapped to custom black-list regions and mitochondria were removed. Two replicates were pooled and MACS2 (*Zhang et al., 2008*) was used to call chromatin opening peaks with options: -f BAMPE -g mm `--broad --broad-cutoff` 0.05. Peaks for all time point were then merged to create nonoverlapping unified peaks. Further data processing was carried out using the R/Bioconductor environment.

For quantitative comparison, HTSeq-count (*Anders et al., 2015*) was used to calculate fragment counts for the nonoverlapping unified peaks and DESeq2 was used to call regions with significantly changed chromatin accessibility. For normalization, ATAC-seq peaks overlapping with promoters of protein-coding genes with unchanged mRNA expression were used to calculate DESeq2 size factors. An adjusted p value of 0.01 was used to identify significantly changed regions by comparing each time point to the 0-hr measurement.

## ChIP-seq data processing

Paired-end ChIP-seq data processing was done as described for ATAC-seq data. For single-end ChIP-seq data, Bowtie2 (*Langmead and Salzberg, 2012*) was used for mapping with '`--local`' option. SAMtools (*Li et al., 2009*) was then used to remove alignments with MAPQ smaller than 7 (-q 7). All published ChIP-seq data were fully processed by ourselves. Detailed information for all ChIP-seq samples can be found in *Supplementary files 4–6*. For peak calling of published paired-end Oct4, Sox2, and Nanog ChIP-seq data, three replicates were pooled and MACS2 (*Zhang et al., 2008*) was used to call peaks with options: -f BAMPE -g mm. For our paired-end Oct4, Sox2 ChIP-seq data, peaks were called by MACS2 with the same options for each replicate. For the H3K4me1 peak used in *Figure 3—figure supplement 1E*, peaks were called by MACS2 with options: -g mm.

For our Oct4 ChIP-seq data, data were normalized using added *D. melanogaster* RNA spike-ins. Normalization factors were obtained by dividing the total *D. melanogaster* spike-ins read counts for each sample by the total spike-ins read counts of the sample with the lowest spike-ins read counts. For our Sox2 ChIP-seq data, data were normalized using a total number of uniquely mapped reads. ChIP-seq coverages were divided by the respective normalization factors. HTSeq-count (*Anders et al., 2015*) was used to calculate fragment counts for peak features. Reads counts were divided by the respective normalization factors.

To avoid noisy signals, we only used Sox2 peaks that were detected by both replicates (consensus peaks) at 0-hr samples for Sox2 occupancy analysis. We overlapped Sox2 consensus peaks with Oct4-sensitive, -insensitive, and -independent enhancers and kept the overlapped peaks for the analysis in *Figures 6C, D and 7C, D*. For Oct4-bound transcribed enhancers this resulted in 154,67 and 111 Sox2 consensus peaks that is overlapped with Oct4-sensitive, -insensitive, and -independent enhancers in *Figure 6C, D*. For Oct4-bound nontranscribed enhancers this resulted in 1,010 and 478 Sox2 consensus peaks that is overlapped with Oct4-sensitive and -insensitive enhancers in *Figure 7C, D*. For Oct4 occupancy analysis in *Figure 7B*, we used the same strategy and this resulted in 844 and 187 Oct4 consensus peaks for Oct4-sensitive and -insensitive enhancers used in *Figure 7B*.

## GO enrichment analysis

The GO enrichment analysis for differentially expressed mRNAs was performed by DAVID Bioinformatics Resources (*Huang et al., 2009*). Genomic regions enrichment analysis was performed by GREAT (*McLean et al., 2010*).

## TF-binding motif analysis

DNA sequences ±500 bp around Oct4 ChIP-seq peak summit were extracted and FIMO (*Grant et al., 2011*) was used to find individual motif occurrences. DNA motifs of Oct4, Sox2, Sox2::Oct4 composite, Klf4, and Esrrb were download from JASPAR database (*Fornes et al., 2020*). Percent of motif occurrence was calculated by counting how many sequences contained the queried motifs in the total of subjected sequences.

## SE annotation in mESC

Previously, 231 large enhancer domains were identified to be SEs from 8794 sites that are co-occupied by Oct4, Sox2, and Nanog in mESC (*Whyte et al., 2013*). We downloaded the annotation from the

original publication and used UCSC LiftOver (*Hinrichs et al., 2006*) to convert the coordinates from mouse mm9 genome assembly to mm10.

## Acknowledgements

We thank Kerstin Maier and Petra Rus for sequencing. LX was supported by the International Max Planck Research School for Genome Science, part of the Göttingen Graduate School for Neurosciences, Biophysics, and Molecular Biosciences. LC was supported by EMBO Long-Term Postdoctoral Fellowship (ALTF-1261-2014). PC was supported by the Deutsche Forschungsgemeinschaft (SFB860, SPP1935, EXC 2067/1-390729940) and the European Research Council (advanced investigator grant CHROMATRANS, grant agreement no. 882357). EAT and KA were supported by funds of the Max Planck Society. SV was supported by the European Research Council (advanced investigator grant (PROMETHEUS, grant agreement no. 669168)), CMM was supported by a White Paper-Project of the Max Planck Society: Brain Organoids -Alternatives to Animal Testing in Neuroscience.

## Additional information

### Funding

| Funder | Grant reference number | Author |
|---|---|---|
| Deutsche Forschungsgemeinschaft | SFB860 | Patrick Cramer |
| Deutsche Forschungsgemeinschaft | SPP1935 | Patrick Cramer |
| Deutsche Forschungsgemeinschaft | EXC 2067/1-390729940 | Patrick Cramer |
| European Research Council | 882357 | Patrick Cramer |
| Max Planck Institute for Multidisciplinary Sciences | open access funding | Patrick Cramer |
| European Research Council | 669168 | Hans R Schöler |
| Max Planck Society | White Paper-project | Hans R Schöler |
| Max Planck Institute for Molecular Biomedicine | Open access funding | Hans R Schöler |

The funders had no role in study design, data collection, and interpretation, or the decision to submit the work for publication.

### Author contributions

Le Xiong, Conceptualization, Formal analysis, Investigation, Methodology, Performed all bioinformatics analysis except footprint analysis during Oct4 recovery, Visualization, Writing – original draft, Writing – review and editing; Erik A Tolen, Conceptualization, Investigation, Methodology, Performed depletion experiments except when stated otherwise, Visualization, Writing – original draft, Writing – review and editing; Jinmi Choi, Investigation, Methodology, Performed TT-seq experiments, Visualization, Writing – review and editing; Sergiy Velychko, Conceptualization, Formal analysis, Investigation, Methodology, Set up the Oct4 recovery experiments, performed ATAC-seq and footprint analysis for Oct4 recovery experiments., Visualization, Writing – original draft, Writing – review and editing; Livia Caizzi, Investigation, Methodology, Performed Oct4 ChIP-seq experiments, Writing – review and editing; Taras Velychko, Methodology, Performed TT-seq for Oct4 recovery experiments., Writing – review and editing; Kenjiro Adachi, Methodology, Writing – review and editing; Caitlin M MacCarthy, Helped with western blots for Oct4 recovery experiment, Writing – review and editing; Michael Lidschreiber, Investigation, Methodology, Visualization, Writing – original draft, Writing – review and editing; Patrick Cramer, Conceptualization, Funding acquisition, Methodology, Project administration, Resources, Supervision, Visualization, Writing – original draft, Writing – review and editing; Hans

R Schöler, Conceptualization, Funding acquisition, Methodology, Project administration, Resources, Supervision, Writing – review and editing

**Author ORCIDs**
Le Xiong (iD) http://orcid.org/0000-0001-8180-768X
Erik A Tolen (iD) http://orcid.org/0000-0002-7147-7506
Sergiy Velychko (iD) http://orcid.org/0000-0002-6227-3966
Livia Caizzi (iD) http://orcid.org/0000-0001-9723-6893
Caitlin M MacCarthy (iD) http://orcid.org/0000-0001-8434-5441
Michael Lidschreiber (iD) http://orcid.org/0000-0002-6740-2755
Patrick Cramer (iD) http://orcid.org/0000-0001-5454-7755
Hans R Schöler (iD) http://orcid.org/0000-0002-7422-8847

**Decision letter and Author response**
Decision letter https://doi.org/10.7554/eLife.71533.sa1
Author response https://doi.org/10.7554/eLife.71533.sa2

## Additional files

### Supplementary files
• Supplementary file 1. Sequencing statistics of TT-seq samples generated in this study, related to *Figure 1*.

• Supplementary file 2. Sequencing statistics of ATAC-seq samples generated in this study, related to *Figure 1*.

• Supplementary file 3. Early and late up- and downregulated gene list, related to *Figure 2*.

• Supplementary file 4. List of previously published ChIP-seq datasets used in this study, related to *Figure 3*.

• Supplementary file 5. Sequencing statistics of Oct4 ChIP-seq samples generated in this study, related to *Figures 4 and 7*.

• Supplementary file 6. Sequencing statistics of Sox2 ChIP-seq samples generated in this study, related to *Figures 6 and 7*.

• Supplementary file 7. Sequencing statistics of TT-seq samples generated in Oct4 recovery experiments, related to *Figure 8*.

• Supplementary file 8. Sequencing statistics of ATAC-seq samples generated in Oct4 recovery experiments, related to *Figure 8*.

• Transparent reporting form

### Data availability
Sequencing data have been deposited in GEO under accession codes GSE174774.

The following dataset was generated:

| Author(s) | Year | Dataset title | Dataset URL | Database and Identifier |
|---|---|---|---|---|
| Xiong L, Choi J, Velychko S, Caizzi L, Velychko T, Adachi K, MacCarthy CM, Lidschreiber M, Cramer P, Schöler HR, Tolen EA | 2021 | Oct4 differentially regulates chromatin opening and enhancer transcription in pluripotent stem cells | https://www.ncbi.nlm.nih.gov/geo/query/acc.cgi?acc=GSE174774 | NCBI Gene Expression Omnibus, GSE174774 |

The following previously published datasets were used:

| Author(s) | Year | Dataset title | Dataset URL | Database and Identifier |
|---|---|---|---|---|
| King HW, Klose RJ | 2017 | The pioneer factor OCT4 requires the chromatin remodeller BRG1 to support gene regulatory element function in mouse embryonic stem cells | https://www.ncbi.nlm.nih.gov/geo/query/acc.cgi?acc=GSE87822 | NCBI Gene Expression Omnibus, GSE87822 |
| Chronis C, Fiziev P, Papp B, Butz S | 2017 | Cooperative binding of Oct4, Sox2, and Klf4 with stage-specific transcription factors orchestrates reprogramming | https://www.ncbi.nlm.nih.gov/geo/query/acc.cgi?acc=GSE90895 | NCBI Gene Expression Omnibus, GSE90895 |

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
