## [Editor Report]

The manuscript by Xiong et al. provides high-resolution kinetic information on transcriptional events and enhancer activity after the loss of the pluripotency factor Oct4. The authors demonstrate different concentration-dependent activities of Oct4 in controlling enhancer transcription and chromatin accessibility. These results are of interest to stem cell biologists and developmental biologists.

---

## [Decision Letter]

**Decision letter after peer review:**

Thank you for submitting your article "Oct4 primarily controls enhancer activity rather than accessibility" for consideration by *eLife*. Your article has been reviewed by 3 peer reviewers, one of whom is a member of our Board of Reviewing Editors, and the evaluation has been overseen by Mone Zaidi as the Senior Editor. The following individual involved in review of your submission has agreed to reveal their identity: David M Suter (Reviewer #3).

Essential revisions:

1) The implications of reduced (rather than absent) Oct4 protein during most of the timecourse of the experiments should be discussed.

2) Control datasets with analysis of enhancers not bound by Oct4, their chromatin accessibility and *Sox2* binding, need to be provided.

3) Report wider expression of pluripotency factors and possible trophectoderm differentiation during Oct4 depletion.

4) Explain limited impact of Oct4 depletion decommissioning super enhancers.

5) The claim that *Sox2* maintains chromatin accessibility requires further experimental support with *Sox2* loss of function.

6) Oct4 recovery experiments followed by nascent transcriptome and chromatin accessibility analysis should be provided to substantiate the claims.

7) Alternative interpretations suggested by the reviewers should be discussed.

*Reviewer #1 (Recommendations for the authors):*

To complete experiments (particularly analysis of regions not bound by Oct4, Oct4 recovery experiments and clarification of the role of *Sox2*), consider alternative explanations and provide a more balanced discussion.

*Reviewer #2 (Recommendations for the authors):*

This study utilizes transcription regulation to achieve depletion of Oct4; this results in a gradual reduction in Oct4 levels, meaning that Oct4 is still present throughout much of the timecourse. On the other hand, Oct4 is only fully removed and absent from chromatin after 12-15 hours, by which time 1000-1500 genes are shown to be differentially regulated. It has previously been described that ESCs can be maintained with low levels of Oct4, resulting in altered signalling requirements and gene expression (Karwacki-Neisius et al. 2013; Radzisheuskaya et al. 2013). Given that Oct4 protein is present at a reduced level for most of the timecourse, the implications of this altered cell state should be discussed.

There are several control datasets that should be described in order to determine how directly loss of Oct4 from enhancers affects their activity. For example, in figure 4, changes in eRNA synthesis at Oct4-bound enhancers are analysed, but there is no corresponding analysis of enhancers not bound by Oct4. Do these remain unchanged as would be expected if the removal of Oct4 from enhancer elements is directly responsible for the reduction in eRNA observed? Similarly, changes in chromatin accessibility are analysed at enhancers bound by Oct4, but it is possible that the changes observed in roughly half of these loci is driven by the change in cellular identity rather than by the loss of Oct4; how does the chromatin accessibility of enhancers not bound by Oct4 change, especially those associated with genes that are downregulated during exit from pluripotency?

It is widely described that loss of Oct4 results in trophectoderm-like differentiation. Was this also seen here?

In Figure 1A, the depletion of Oct4 protein would benefit from quantification.

Figure 1 shows that *Sox2* and Nanog protein levels are stable for some time during Oct4 depletion. It would be useful to see how stable the expression of a wider array of pluripotency-associated factors, for example from the TT-seq data.

In Figure 4G, super enhancers which are bound by Oct4 and show a decrease in enhancer RNA, and whose nearest gene shows a decrease in expression, are analysed and this indicates that mRNA expression decrease is a subsequent event to eRNA downregulation. However, around half of the super enhancer elements bound by Oct4 with a decrease in enhancer RNA do not show a decrease in neighbouring gene expression. Is this simply the result of these enhancers regulating more distant targets? If not, how do the authors explain the limited impact of decommissioning these super enhancers?

Regarding the transient maintenance of *Sox2* following removal of Oct4, this is reminiscent of similar findings relating to the continued binding of Nanog in the absence of Oct4 recently reported (Bates et al. 2021). This paper also described enhancer decommissioning and retained transcription factor binding, as well as very limited analysis of loss of enhancer RNA expression, and should be referenced.

In figures 4, 5, and 6 a similar analysis of enhancer RNA expression, chromatin accessibility, and *Sox2* binding at enhancers not bound by Oct4 should be performed in order to support that loss of Oct4 is directly responsible for their downregulation, or to indicate that it may be the result of indirect effects.

Throughout the paper, the number of replicates should be added to figure legends.

In the discussion, it is stated that Oct4 mainly acts as an activator whereas *Sox2* maintains chromatin accessibility. Regarding Oct4 acting as an activator, this appears to be contradicted by the finding that even within 6 hours of adding doxycycline, equal numbers of genes are up and downregulated (figure 2a); while it appears that Oct4 acts as an activator of eRNAs (figure 4a), it is not clear that it specifically acts as an activator of gene expression more broadly. The implication that *Sox2* maintains chromatin accessibility seems highly speculative. It is possible that many other factors remain bound for some time following loss of Oct4, and any or a cumulative effect of these may result in the maintenance of chromatin accessibility.

In support of the role for *Sox2* in maintaining open chromatin and allowing Oct4 to bind, Chen et al. 2014 is cited (note, this reference is not found in the reference list at the end of the manuscript). However, reanalysis of these data by Biddle et al. (2019) indicates that the analysis performed by Chen et al. is overly simplified, and in truth the effect of Oct4 and *Sox2* on one-another's binding is likely to vary across different binding sites. As such, the reanalysis does not support the sequential binding put forth by Chen et al.

*Reviewer #3 (Recommendations for the authors):*

My suggestion to the authors is to re-interpret their datasets, and also to complete their datasets with Oct4 recovery experiments followed by nascent transcriptome and chromatin accessibility analysis.

[Editors' note: further revisions were suggested prior to acceptance, as described below.]

Thank you for resubmitting your work entitled "Oct4 differentially regulates chromatin opening and enhancer transcription in pluripotent stem cells" for further consideration by *eLife*. Your revised article has been evaluated by Mone Zaidi (Senior Editor) and a Reviewing Editor in consultation with the previous reviewers.

We all agree that the manuscript has been largely improved and is potentially acceptable for publication. However, following your e-mail explaining that some cells tested positive for mycoplasma and that this should not affect the conclusions of your study, we believe that these explanations should be discussed in the main manuscript as well, for clarity and transparency.

Please add the arguments below, suggested in your correspondence, to the main text:

1) The ZHBTc4 mESC cells exhibit normal growth patterns and morphology. Figure 1 supplement figure 1a confirms that cell morphology was normal.

2) Our experiments were all done in relation to the untreated ZHBTc4 cells. Therefore, we do not make skewed comparisons but evaluate the collected results and the resulting conclusions in a consistent manner.

3) Our results confirm previously published observations with ZHBTc4 mESCs that were mycoplasma negative. For example, the observation that loss of chromatin accessibility is linked to the loss of Oct4 was also observed in two other publications. Namely, Friman et al. 2019 *eLife* and King and Klose 2017 *eLife*.

4) Finally, we found that depletion of Oct4 leads to rapid down-regulation of key pluripotency factors. These findings are in line with experiments from Bates et al. 2021 Stem Cell Reports, in which a degron system was utilized to deplete Oct4 protein. We thoroughly compared immediate gene expression changes upon Oct4 depletion observed in our study and in Bates et al. and found similar changes (see response letter, author response figures 1 and 6).

---

## [Author Response]

Essential revisions:1) The implications of reduced (rather than absent) Oct4 protein during most of the timecourse of the experiments should be discussed.

We agree that this needs to be discussed. As mentioned by reviewer 2 it is correct that in our Tet-off system (Niwa et al., 2000) Oct4 is gradually depleted and only fully removed and absent from chromatin after 12-15 hours (h) of DOX treatment. In the entire manuscript we replaced "rapid depletion" by either "gradual depletion" or just "depletion".

The gradual depletion provided insights into Oct4’s concentration-dependent role in gene expression regulation. Enhancer and target gene transcription were very sensitive to even small reductions of Oct4 binding, whereas chromatin accessibility was either unaffected (Oct4-insensitive enhancers) or affected later when Oct4 levels decreased considerably (Oct4-sensitive enhancers). During revision we conducted the proposed Oct4 recovery experiments (see Essential revisions point 6) and the results provide strong evidence to support that low Oct4 levels are sufficient to maintain chromatin accessibility at Oct4-sensitive enhancers, in agreement with the alternative model proposed by reviewer 3. Thus, in light of the added recovery experiments and reviewer comments, we reinterpreted some of our results and edited the Discussion section of the manuscript accordingly (see Essential revisions points 6 and 7 for more details). We acknowledge also that it has previously been described that ESCs can be maintained with low levels of Oct4, resulting in altered signalling requirements and gene expression (Karwacki-Neisius et al. 2013; Radzisheuskaya et al. 2013). We cite the literatures mentioned by the reviewer, including also Bates et al. 2021.

Briefly, in Bates et al. 2021, Oct4 protein was fully depleted within 2 h of IAA addition. Using RNA-seq, the authors showed that full depletion of Oct4 leads to rapid down-regulation of key pluripotency factors, followed by up-regulation of several trophoblast- associated genes (Figure 2B, Bates et al. 2021). Moreover, the authors also showed by RNA-seq that when using conventional tamoxifen-induced CreER-driven genetic ablation of the Oct4 system, there is no influence on gene expression of key pluripotency factors after 24 and 36 h depletion of Oct4 (Figure 2A, Bates et al. 2021), by which time Oct4 expression was nearly completely abolished. In contrast, in our study, we used a Tet-off Oct4 depletion system that leads to an exit of pluripotency when the Oct4 level is reduced by over 50% (Niwa et al. 2000). Importantly, using TT-seq, we found rapid down-regulation of well-known key pluripotency factors (Figure 2H, see also Essential Revisions point 3 for selected pluripotency factors), which is consistent with Bates et al. 2021 (Figure 2B). Thus, although Oct4 was depleted in different ways, both studies support the same conclusion that loss of Oct4 results in rapid down-regulation of pluripotency factors. In our study, we used TT-seq, which is more sensitive than RNA-seq for detection of changes in gene transcription upon perturbation (Schwalb et al. 2016, Michel et al. 2017).

We went on to further compare gene expression changes observed in Bates et al., 2021 with mRNA synthesis changes observed in our system. In our study, Oct4 was gradually depleted and we identified ~1500 differentially regulated genes and classified them into early and late down- and up-regulated genes based on their kinetics (Figure 2A-D). In addition, we found that super enhancer (SE)-controlled genes were particularly sensitive to Oct4 depletion (Figure 2F-H). To investigate whether similar effects can be observed in the Oct4 protein depletion system from Bates et al., we analyzed the published RNA-seq data. As shown in Author response image 1, similar effects can also be seen in the RNA-seq data taken from Bates et al. 2021: (1) early down-regulated genes showed a stronger (earlier) decrease of mRNA synthesis/gene expression compared to late down-regulated genes in both data sets (A: left, TT-seq data, this study; right, RNA-seq data, Bates et al.), although the differences are diminished after prolonged (24 h) Oct4 protein depletion in Bates et al. data; (2) early and late up-regulated genes also behaved similarly in both data sets (B: left, TT-seq data, this study; right, RNA-seq data, Bates et al.); (3) we also found SE-controlled genes were more sensitive to Oct4 protein depletion in the data taken from Bates et al. (C), consistent with our TT-seq data shown in Figure 2G. Taken together, the overall changes in mRNA synthesis/gene expression and their kinetics are comparable between our gradual Oct4 depletion system and the full Oct4 protein depletion after 2 h published in Bates et al., 2021.

**Author response image 1. sa2fig1:** Response kinetics of different types of differentially regulated gene groups in our TT-seq data and RNA-seq data from Bates et al. , 2021. (A) Response kinetics of early/late down-regulated genes. (B) Response kinetics of early/late up-regulated genes. (C) Responses kinetics of SE nearest down-regulated genes and other down-regulated genes using RNA-seq data from Bates et al., 2021, in comparison to Figure 2G.

2) Control datasets with analysis of enhancers not bound by Oct4, their chromatin accessibility and Sox2 binding, need to be provided.

We agree that these are valuable controls. We now provide the proposed control data sets and the corresponding analysis of enhancers not bound by Oct4 (Oct4-unbound transcribed enhancers; see Figure 3A), their eRNA synthesis, accessibility and *Sox2* binding changes. Briefly, we performed differential expression analysis of eRNAs for Oct4-unbound transcribed enhancers (Figure 4—figure supplement2A-E). Based on these results, we also generated a barplot to compare the percentage of differentially regulated eRNAs at each time point for Oct4-bound and -unbound transcribed enhancers, respectively (Figure 4—figure supplement 2F). We found that down-regulation of eRNA synthesis occurred predominately at Oct4-bound transcribed enhancers in comparison to Oct4-unbound transcribed enhancers (Figure 4—figure supplement 2F, left). Although a total of ~400 eRNAs (7.4%) were down-regulated at Oct4-unbound transcribed enhancers, most of the downregulation occurred late, between 12-15 h of Oct4 depletion. A tiny amount of up-regulated eRNAs (<2%) was detected at both Oct4-bound and -unbound transcribed enhancers (Figure 4—figure supplement 2F, right). These data suggest that upon Oct4 depletion, inactivation of enhancers occurred rapidly and predominately at Oct4-bound transcribed sites, although a small fraction of Oct4-unbound transcribed sites was also influenced later, most likely representing secondary effects. This figure is now described in the subsection “Oct4 is often required for enhancer transcription”.

We also called significantly changed accessible chromatin regions for Oct4-unbound transcribed enhancers (Figure 5—figure supplement 2). After 15 h of DOX treatment only 48 enhancers (0.6%) showed a significant change in chromatin accessibility, indicating that chromatin accessibility remained largely unaltered across the full time course at Oct4-unbound transcribed enhancers. Figure 5—figure supplement 2 is now described in the subsection “Reduced Oct4 binding does not immediately influence chromatin accessibility”.

Finally, we also investigated *Sox2* occupancy at Oct4-unbound transcribed enhancers. Only 307 (4.3%) of Oct4-unbound transcribed enhancers overlapped with our *Sox2* ChIP-seq peaks (within a 1kb window). For this subgroup, we also found no appreciable change in *Sox2* occupancy across the full time course (Figure 6—figure supplement 3). This figure is now addressed in the subsection “*Sox2* remains transiently bound at Oct4-sensitive enhancers during Oct4 depletion”.

3) Report wider expression of pluripotency factors and possible trophectoderm differentiation during Oct4 depletion.

In Figure 2 we show that pluripotency genes were early down-regulated upon Oct4 depletion. Moreover, we have provided early and late down- and up-regulated gene lists as supplementary table (Table S3). In addition, we now also provide the corresponding normalized TT-seq count RPK tables as resource table (Figure 2-source data 1). To further illustrate the down-regulation of pluripotency factors and up-regulation of genes associated with trophectoderm differentiation, we plotted mRNA synthesis changes over time as line plots (Author response image 2):

**Author response image 2. sa2fig2:** mRNA synthesis changes for pluripotency factors and genes associated with trophectoderm differentiation over time. (A) Genes coding for pluripotency factors (B) Genes associated with trophectoderm differentiation. Y axis represents TT-seq read counts in reads per kilobase (RPKs).

Finally, Bates et al. 2021 found that up-regulation of genes associated with trophectoderm differentiation occurred after down-regulation of pluripotency markers, suggesting exit from pluripotency precedes trophectoderm gene expression upon loss of Oct4. Based on the mRNA synthesis changes of pluripotency factors and trophectoderm associated genes (Author response image 2), we generated a corresponding fold change scatterplot to compare the dynamics of their synthesis changes over time. We observed a decrease of pluripotency gene expression already at 3 and 6 h of DOX treatment, while expression of trophectoderm related genes started to increase later, at 6 and 9 h (Author response image 3). Thus, our data also suggests that exit of pluripotency precedes trophectoderm differentiati­­­on.

**Author response image 3. sa2fig3:** Exit of pluripotency precedes trophectoderm differentiation. Pluripotency genes (red) correspond to genes plotted in Author response image 2, trophectoderm genes (green) correspond to genes plotted in Author response image 2. Medians for each gene group at each time point are represented by black triangle. Y axis indicates log2 fold change relative to 0h.

4) Explain limited impact of Oct4 depletion decommissioning super enhancers.

The second reviewer noted correctly that for around half of the super enhancer (SE) elements bound by Oct4 a decrease in eRNA synthesis did not lead to a statistically significant decrease of nearest gene expression. To explore this further, we first plotted SE nearest gene mRNA synthesis fold changes to investigate whether we could detect small differences in fold changes that may simply not have exceeded our statistical significance level. This analysis also revealed no appreciable changes in SE nearest gene mRNA synthesis (Author response image 4). Next, we analyzed the occupancy of Oct4 and the fold change of eRNA synthesis at these SEs. We found significantly lower occupancy of Oct4 (Figure 4—figure supplement 3, left) and significantly lower decrease of eRNA synthesis (Figure 4—figure supplement 3, right) at these SEs in comparison to the decommissioned SEs for which nearest genes were down-regulated. This suggests that Oct4 occupancy and the degree of eRNA synthesis changes at transcriptionally down-regulated SEs may play a role in transcription. Figure 4—figure supplement 3 is now also discussed in the subsection “Oct4 binds enhancers to activate putative target genes”.

**Author response image 4. sa2fig4:** Characterization of SE-gene pairs for which nearest gene transcription remained unchanged. (A) Boxplots showing mRNA synthesis fold change for unchanged nearest genes of decommissioned SEs. (B) Boxplot showing genomic distances from the decommissioned SEs for which nearest gene transcription remained unchanged to the nearest down-regulated genes detected in TT-seq.

Previously, it has been shown that even after 24 h of Oct4 depletion, only for a subset (23%, 1625/7163) of Oct4-occupied enhancers that experienced a decrease of chromatin accessibility, down-regulation of nearest genes was observed (King and Klose, 2017, Figure 2F). Together with our results this suggests that decommissioning of enhancers does not always lead to down-regulation of gene expression. It is known that regulation by enhancers is not limited to the nearest gene and currently it is unclear whether the decommissioned SEs may have regulatory ability beyond the nearest gene. To explore this possibility, we analyzed the decommissioned SEs for which nearest gene transcription was unchanged and calculated their genomic distances to the nearest down-regulated genes detected in TT-seq. This analysis revealed a rather large median distance of 674 kb (Author response image 4). We think this indicates that the majority of these SEs do not regulate genes beyond the nearest one, although it can’t be excluded that long range enhancer-promoter interactions may exist.

5) The claim that Sox2 maintains chromatin accessibility requires further experimental support with Sox2 loss of function.

We agree that the claim that *Sox2* maintains chromatin accessibility would require further experimental support. Currently there is no rapid *Sox2* depletion system. Using the same Tet-off system as we used for Oct4 depletion, Friman et al. 2019 showed that it took 26-40 h to achieve complete depletion of *Sox2*. The development of a rapid *Sox2* protein depletion system will be crucial to dissect *Sox2*’s function in chromatin accessibility in the future, but is beyond the scope of the current work.

We conducted the proposed Oct4 recovery experiments (see Essential Revisions point 6) and the results provide strong evidence to support that low Oct4 levels, rather than *Sox2*, are sufficient to maintain chromatin accessibility at Oct4-sensitive enhancers, in agreement with the alternative model proposed by reviewer 3. Thus, in light of the added recovery experiments and reviewer comments, we reinterpreted some of our results. The claims regarding *Sox2* transiently maintaining chromatin accessibility upon Oct4 loss at Oct4-sensitive enhancers were removed from the revised manuscript.

E.g., we changed the following in the Results section:

– revised text, p.8 l.302: "*Sox2* may contribute to retained enhancer accessibility" to "*Sox2* remains transiently bound at Oct4-sensitive enhancers during Oct4 depletion".

– revised text, p.9 l.342: "these findings suggest that *Sox2* can maintain chromatin accessibility" to "these findings reveal that during Oct4 depletion *Sox2* binding and chromatin accessibility change with similar kinetics at Oct4-bound enhancers."

Furthermore, we removed the corresponding interpretations of the data from the Discussion section, e.g.:

"Our results suggest that Oct4 mainly acts as an activator that stimulates transcription of pluripotency enhancers and their target genes, whereas *Sox2* acts as a factor that renders chromatin accessible."

"*Sox2* can retain enhancer accessibility for some time after Oct4 depletion."

"In conclusion, our data indicate that the primary function of Oct4 is to control enhancer activity rather than accessibility."

"Thus, a general model emerges that Oct4 controls enhancer activity, whereas *Sox2* governs enhancer accessibility, …"

And instead, the following paragraph was added to the Discussion section:

"… at enhancers dependent on Oct4 for their accessibility (Oct4-sensitive enhancers) *Sox2* remained transiently bound during the first 6 h of DOX treatment and only decreased when chromatin accessibility also decreased. This can be explained by the proposed model that low levels of Oct4 are sufficient to retain chromatin accessible, thus leading to a loss of *Sox2* binding only when Oct4 levels decreased to the point where chromatin accessibility can no longer be maintained. This befits the pioneer role of Oct4 in creating an open chromatin environment to support binding of other pluripotency factors (Friman et al., 2019; King and Klose, 2017). In addition, reduced levels of Oct4 may cooperate with *Sox2* to temporarily retain enhancers accessible. Furthermore, we found enhanced binding of *Sox2* at Oct4-bound enhancers after 3 h of DOX treatment. This suggests that Oct4 may inhibit *Sox2* binding at normal levels in ESCs, and agrees with a study that has shown that overexpression of Oct4 decreases *Sox2* binding (Biddle et al., 2019). This is similar to a recent study in which enhanced binding of Nanog was observed after complete degradation of Oct4 protein (Bates et al., 2021)."

6) Oct4 recovery experiments followed by nascent transcriptome and chromatin accessibility analysis should be provided to substantiate the claims.

We thank the reviewers for the suggestion and we performed the proposed Oct4 recovery experiments. We added these findings as a new results subsection “Partial recovery of Oct4 increases chromatin accessibility and enhancer transcription”. Please refer to the revised manuscript for details.

7) Alternative interpretations suggested by the reviewers should be discussed.

We think the Oct4 recovery experiments (see Essential Revisions point 6) provide strong evidence to support that low levels of Oct4 are sufficient to regulate chromatin accessibility, in agreement with the alternative model proposed by reviewer 3. Thus, in light of the added recovery experiments and reviewer comments, we reinterpreted some of our results and edited the discussion accordingly. E.g., we added the following paragraphs in the Discussion section of the revised manuscript:

"Here we used Oct4 depletion from mouse ESCs and several genomic approaches to analyze the role of Oct4 in the maintenance of pluripotency over a high-resolution time course. We used a Tet-off system that allows gradual Oct4 depletion (Niwa et al., 2000), within 15h. The gradual depletion provided insights into Oct4’s concentration-dependent role in gene expression regulation. Our data showed that loss of Oct4 from enhancers goes along with a decrease in mRNA synthesis from Oct4 target genes crucial for maintaining pluripotency.

During Oct4 depletion, enhancer and target gene transcription were very sensitive to even small reductions of Oct4 binding, whereas chromatin accessibility was either unaffected (Oct4-insensitive enhancers) or affected later when Oct4 levels decreased considerably (Oct4-sensitive enhancers). During partial recovery of Oct4, chromatin accessibility and enhancer transcription increased at Oct4-sensitive enhancers, but eRNA synthesis could not be fully reactivated, although chromatin accessibility was largely restored. Together, this indicates that normal ESC levels of Oct4 are required for enhancer transcription whereas low levels of Oct4 are sufficient to retain chromatin accessibility. This is also consistent with a recent study, in which the authors used an Oct4-low expressed protein depletion system in ESCs and observed that loss of chromatin accessibility was quasi-synchronized with Oct4 loss (Friman et al., 2019)."

"… at enhancers dependent on Oct4 for their accessibility (Oct4-sensitive enhancers) *Sox2* remained transiently bound during the first 6 h of DOX treatment and only decreased when chromatin accessibility also decreased. This can be explained by the proposed model that low levels of Oct4 are sufficient to retain chromatin accessible, thus leading to a loss of *Sox2* binding only when Oct4 levels decreased to the point where chromatin accessibility can no longer be maintained. This befits the pioneer role of Oct4 in creating an open chromatin environment to support binding of other pluripotency factors (Friman et al., 2019; King and Klose, 2017). In addition, reduced levels of Oct4 may cooperate with *Sox2* to temporarily retain enhancers accessible. Furthermore, we found enhanced binding of *Sox2* at Oct4-bound enhancers after 3 h of DOX treatment. This suggests that Oct4 may inhibit *Sox2* binding at normal levels in ESCs, and agrees with a study that has shown that overexpression of Oct4 decreases *Sox2* binding (Biddle et al., 2019). This is similar to a recent study in which enhanced binding of Nanog was observed after complete degradation of Oct4 protein (Bates et al., 2021)."

Furthermore, we removed some of the previous interpretations of the data from the Discussion section, e.g.:

"Our results suggest that Oct4 mainly acts as an activator that stimulates transcription of pluripotency enhancers and their target genes, whereas *Sox2* acts as a factor that renders chromatin accessible."

"*Sox2* can retain enhancer accessibility for some time after Oct4 depletion."

"In conclusion, our data indicate that the primary function of Oct4 is to control enhancer activity rather than accessibility."

"Thus, a general model emerges that Oct4 controls enhancer activity, whereas *Sox2* governs enhancer accessibility, …"

Taken together the results from the Oct4 depletion and recovery experiments the reinterpretation of the data can be summarized as follows:

"In summary, we investigated the role of Oct4 in regulating chromatin accessibility and transcriptional activity in pluripotent cells using high resolution time course data for Oct4 depletion and subsequent recovery. We discovered that Oct4 has differential concentration-dependent activities in regulating enhancer transcription and chromatin accessibility in pluripotent stem cells. Normal ESC concentrations of Oct4 are required to maintain the transcription of pluripotency enhancers. In contrast, Oct4 regulates chromatin accessibility very rapidly even at low concentrations, capable of increasing chromatin accessibility within hours of induction."

Finally, we have updated the manuscript title from "Oct4 primarily controls enhancer activity rather than accessibility" to "Oct4 differentially regulates chromatin opening and enhancer transcription in pluripotent stem cells" and edited the abstract of the manuscript accordingly.

Reviewer #1 (Recommendations for the authors):To complete experiments (particularly analysis of regions not bound by Oct4, Oct4 recovery experiments and clarification of the role of Sox2), consider alternative explanations and provide a more balanced discussion.

We thank the reviewer for the critical review and the suggestions that helped us to improve the manuscript. We have now also analyzed regions not bound by Oct4 as seen in the response to point 2 in Essential revisions. Moreover, we have performed Oct4 recovery experiments as seen in the response to point 6 in Essential revisions. In light of the added recovery experiments and reviewer comments, we reinterpreted some of our results, clarified the role of *Sox2*, and edited the Discussion section of the manuscript accordingly (see Essential revisions points 5 and 7 for more details).

Reviewer #2 (Recommendations for the authors):This study utilizes transcription regulation to achieve depletion of Oct4; this results in a gradual reduction in Oct4 levels, meaning that Oct4 is still present throughout much of the timecourse. On the other hand, Oct4 is only fully removed and absent from chromatin after 12-15 hours, by which time 1000-1500 genes are shown to be differentially regulated. It has previously been described that ESCs can be maintained with low levels of Oct4, resulting in altered signalling requirements and gene expression (Karwacki-Neisius et al. 2013; Radzisheuskaya et al. 2013). Given that Oct4 protein is present at a reduced level for most of the timecourse, the implications of this altered cell state should be discussed.There are several control datasets that should be described in order to determine how directly loss of Oct4 from enhancers affects their activity. For example, in figure 4, changes in eRNA synthesis at Oct4-bound enhancers are analysed, but there is no corresponding analysis of enhancers not bound by Oct4. Do these remain unchanged as would be expected if the removal of Oct4 from enhancer elements is directly responsible for the reduction in eRNA observed? Similarly, changes in chromatin accessibility are analysed at enhancers bound by Oct4, but it is possible that the changes observed in roughly half of these loci is driven by the change in cellular identity rather than by the loss of Oct4; how does the chromatin accessibility of enhancers not bound by Oct4 change, especially those associated with genes that are downregulated during exit from pluripotency?

We thank the reviewer for the critical review and the insightful suggestions that helped us to improve the manuscript. We have addressed these questions, as seen in our responses to points 1 and 2 in Essential revisions.

It is widely described that loss of Oct4 results in trophectoderm-like differentiation. Was this also seen here?

See response to point 3 in Essential revisions.

In Figure 1A, the depletion of Oct4 protein would benefit from quantification.

We did quantification for both whole cell lysates and chromatin fraction western blots and added them as resource data of figure 1A. Moreover, quantification of Oct4 and *Sox2* levels in the chromatin fraction were added to figure legend 1A.

Figure 1 shows that Sox2 and Nanog protein levels are stable for some time during Oct4 depletion. It would be useful to see how stable the expression of a wider array of pluripotency-associated factors, for example from the TT-seq data.

See response to point 3 in Essential revisions.

In Figure 4G, super enhancers which are bound by Oct4 and show a decrease in enhancer RNA, and whose nearest gene shows a decrease in expression, are analysed and this indicates that mRNA expression decrease is a subsequent event to eRNA downregulation. However, around half of the super enhancer elements bound by Oct4 with a decrease in enhancer RNA do not show a decrease in neighbouring gene expression. Is this simply the result of these enhancers regulating more distant targets? If not, how do the authors explain the limited impact of decommissioning these super enhancers?

See response to point 4 in Essential revisions.

Regarding the transient maintenance of Sox2 following removal of Oct4, this is reminiscent of similar findings relating to the continued binding of Nanog in the absence of Oct4 recently reported (Bates et al. 2021). This paper also described enhancer decommissioning and retained transcription factor binding, as well as very limited analysis of loss of enhancer RNA expression, and should be referenced.

We have added this reference and we absolutely agree that the observation for *Sox2* is similar to the findings for Nanog in Bates et al. 2021. We have added one sentence in to the Discussion section: “This is similar to a recent study in which enhanced binding of Nanog was observed after complete degradation of Oct4 protein (Bates et al., 2021)”

For some comparative analysis between our data and the data from Bates et al. 2021 see responses to points 1 and 3 in Essential Revisions.

In figures 4, 5, and 6 a similar analysis of enhancer RNA expression, chromatin accessibility, and Sox2 binding at enhancers not bound by Oct4 should be performed in order to support that loss of Oct4 is directly responsible for their downregulation, or to indicate that it may be the result of indirect effects.

See response to point 2 in Essential revisions.

Throughout the paper, the number of replicates should be added to figure legends.

Two replicates were used throughout the paper and we have added the number of replicates in the figure legends.

In the discussion, it is stated that Oct4 mainly acts as an activator whereas Sox2 maintains chromatin accessibility. Regarding Oct4 acting as an activator, this appears to be contradicted by the finding that even within 6 hours of adding doxycycline, equal numbers of genes are up and downregulated (figure 2a); while it appears that Oct4 acts as an activator of eRNAs (figure 4a), it is not clear that it specifically acts as an activator of gene expression more broadly. The implication that Sox2 maintains chromatin accessibility seems highly speculative. It is possible that many other factors remain bound for some time following loss of Oct4, and any or a cumulative effect of these may result in the maintenance of chromatin accessibility.

We agree that it is not clear that Oct4 specifically acts as an activator of gene expression more broadly. We have modified the text in the discussion and only emphasize that Oct4 acts as an activator for pluripotency genes.

We also agree with the reviewer’s comment that the implication that *Sox2* maintains chromatin accessibility seems highly speculative, for details see response to point 5 in Essential Revisions.

In support of the role for Sox2 in maintaining open chromatin and allowing Oct4 to bind, Chen et al. 2014 is cited (note, this reference is not found in the reference list at the end of the manuscript). However, reanalysis of these data by Biddle et al. (2019) indicates that the analysis performed by Chen et al. is overly simplified, and in truth the effect of Oct4 and Sox2 on one-another's binding is likely to vary across different binding sites. As such, the reanalysis does not support the sequential binding put forth by Chen et al.

We thank the reviewer for correcting this and we have removed this reference from the discussion.

Reviewer #3 (Recommendations for the authors):My suggestion to the authors is to re-interpret their datasets, and also to complete their datasets with Oct4 recovery experiments followed by nascent transcriptome and chromatin accessibility analysis.

We thank the reviewer for the suggestions. We performed the proposed Oct4 recovery experiments and based on the results and in the light of the reviewer comments we have reinterpreted some of our data sets and modified the manuscript accordingly. See responses to points 5 – 7 in Essential revisions.

[Editors' note: further revisions were suggested prior to acceptance, as described below.]

We all agree that the manuscript has been largely improved and is potentially acceptable for publication. However, following your e-mail explaining that some cells tested positive for mycoplasma and that this should not affect the conclusions of your study, we believe that these explanations should be discussed in the main manuscript as well, for clarity and transparency.Please add the arguments below, suggested in your correspondence, to the main text:1) The ZHBTc4 mESC cells exhibit normal growth patterns and morphology. Figure 1 supplement figure 1a confirms that cell morphology was normal.2) Our experiments were all done in relation to the untreated ZHBTc4 cells. Therefore, we do not make skewed comparisons but evaluate the collected results and the resulting conclusions in a consistent manner.3) Our results confirm previously published observations with ZHBTc4 mESCs that were mycoplasma negative. For example, the observation that loss of chromatin accessibility is linked to the loss of Oct4 was also observed in two other publications. Namely, Friman et al. 2019 eLife and King and Klose 2017 eLife.4) Finally, we found that depletion of Oct4 leads to rapid down-regulation of key pluripotency factors. These findings are in line with experiments from Bates et al. 2021 Stem Cell Reports, in which a degron system was utilized to deplete Oct4 protein. We thoroughly compared immediate gene expression changes upon Oct4 depletion observed in our study and in Bates et al. and found similar changes (see response letter, author response figures 1 and 6).

We thank the reviewers and the editors for evaluating the revised manuscript. We absolutely agree that the mycoplasma contamination should be openly discussed in the main manuscript for clarity and transparency. We have now added the following paragraph to the Discussion section of the manuscript: "Analysis of the TT-seq data revealed that the used ZHBTc4 cells were contaminated with mycoplasma (Methods). However, this does not alter the conclusions described in this manuscript. First, the ZHBTc4 cells exhibit normal growth patterns and morphology (Figure 1—figure supplement 1A). Second, our experiments and analyses were all done in relation to the untreated ZHBTc4 cells. Third, as discussed, our results confirm previously published observations with mycoplasma-negative mESCs (Friman et al., 2019; King and Klose, 2017; Bates et al., 2021)."

Moreover, for a more detailed comparison of immediate gene expression changes upon Oct4 depletion observed in our study and in Bates et al., 2021, see author response figures 1 and 6 in Essential Revisions. Similar changes in gene expression were observed.

Finally, we edited the methods section and state now also the percentage of TT-seq reads that mapped to mycoplasma (~0.3%).